# Controllable and Decomposed Diffusion Models for Structure-based Molecular Optimization

**Xiangxin Zhou**[1,2,3*] **Xiwei Cheng**[3,4*] **Yuwei Yang**[3] **Yu Bao**[3] **Liang Wang**[1,2] **Quanquan Gu**[3†]

[1]School of Artificial Intelligence, University of Chinese Academy of Sciences
[2]Center for Research on Intelligent Perception and Computing (CRIPAC),
 State Key Laboratory of Multimodal Artificial Intelligence Systems (MAIS),
 Institute of Automation, Chinese Academy of Sciences (CASIA)
[3]ByteDance Research
[4]Halıcıoğlu Data Science Institute, University of California San Diego

## Abstract

Recently, 3D generative models have shown promising performances in structure-based drug design by learning to generate ligands given target binding sites. However, only modeling the target-ligand distribution can hardly fulfill one of the main goals in drug discovery – designing novel ligands with desired properties, e.g., high binding affinity, easily synthesizable, etc. This challenge becomes particularly pronounced when the target-ligand pairs used for training do not align with these desired properties. Moreover, most existing methods aim at solving *de novo* design task, while many generative scenarios requiring flexible controllability, such as R-group optimization and scaffold hopping, have received little attention. In this work, we propose DecompOpt, a structure-based molecular optimization method based on a controllable and decomposed diffusion model. DecompOpt presents a new generation paradigm which combines optimization with conditional diffusion models to achieve desired properties while adhering to the molecular grammar. Additionally, DecompOpt offers a unified framework covering both *de novo* design and controllable generation. To achieve so, ligands are decomposed into substructures which allows fine-grained control and local optimization. Experiments show that DecompOpt can efficiently generate molecules with improved properties than strong *de novo* baselines, and demonstrate great potential in controllable generation tasks.

## 1 Introduction

Structure-based drug design (SBDD) (Anderson, 2003) is an approach that involves designing drug molecules based on the 3D structure of a target. The goal of SBDD is to generate ligands with desired properties which can bind tightly to the target binding site. Recently, several works have cast SBDD into a conditional 3D molecular generation task and achieved remarkable success thanks to the powerful deep generative models. In these models, the target binding site serves as the condition and the conditional distribution of ligands is learnt in a data-driven manner using various generative models. Peng et al. (2022) and Zhang et al. (2022) proposed to generate molecules given pockets in an auto-regressive fasion using atoms or fragments as building blocks respec-

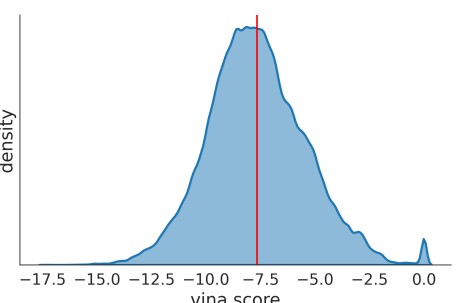

Figure 1: Vina Scores distribution of protein-ligand pairs in CrossDocked2020 dataset. $-8.18$ kcal/mol, marked by the red vertical line, is a commonly used value representing moderate binding affinity.

---

*Equal Contribution. Work was done during Xiangxin's and Xiwei's internship at ByteDance.
†Corresponding Author: Quanquan Gu (quanquan.gu@bytedance.com).

tively. Guan et al. (2023a); Schneuing et al. (2022); Lin et al. (2022) use diffusion models to generate ligands by modeling atom types and positions.

Generative models are powerful approaches for extracting the underlying molecular grammar (e.g., the reasonable atomic valence, stable molecular conformation, etc.). However, they cannot generate molecules with desired properties if the training data do not align with these properties as well. Indeed, unsatisfying data quality is a common challenges in drug discovery (Vamathevan et al., 2019). As Figure 1 shows, the ligands in CrossDocked2020 (Francoeur et al., 2020), a widely used training dataset for SBDD models, have moderate binding affinities measured by molecular docking scores. Solely maximizing the likelihood of training data can mislead SBDD models and cause inefficiency in generating potential drug candidates. To overcome this limitation, molecular optimization (Xie et al., 2021; Fu et al., 2022) offers a direct path for searching molecules with desired properties in the broad chemical space. However, its application to 3D molecule generation remains unexplored.

On the other hand, current SBDD models are mostly limited to *De novo* design (Hartenfeller & Schneider, 2011), which focuses on generating ligands from scratch, is the main task that most efforts have been devoted to. However, controllable molecular generation scenarios, such as R-group design (Takeuchi et al., 2021) (also known as biosteric replacement) and scaffold hopping (Böhm et al., 2004), are equally, if not more, important. Unlike *de novo* design, controllable generation tasks start from an existing compound, and only modify a local substructure to improve the synthetic accessibility, potency and drug-likeness properties or to move into novel chemical space for patenting (Langdon et al., 2010). Controllable generation aims to utilize prior knowledge, such as a known active compound, in the design process to increase the chance of finding promising candidates. Some initial efforts have been made to address controllable molecular generation problem. For example, Igashov et al. (2022); Imrie et al. (2020; 2021); Huang et al. (2022) propose to use generative models to design linkers between the given fragments. However, these methods are designed for a specific controllable generation task and cannot be generalized.

To overcome the aforementioned challenges and limitations in existing SBDD approaches, we propose DECOMPOPT, a controllable and decomposed diffusion model for structure-based molecular optimization. DECOMPOPT combines diffusion models with optimization algorithm to harness the advantages of both approaches. Diffusion models are used to extract molecular grammar in a data-driven fashion, while optimization algorithm is used to effectively optimize the desired properties. Furthermore, DECOMPOPT offers a unified generation framework for both *de novo* design and controllable generation through ligands decomposition. Notably, a ligand that binds to a target binding site can be naturally decomposed into several substructures, i.e., arms and scaffold, where arms locally interact with corresponding subpockets and scaffold links all arms to form a complete molecule. Such decomposition motivates us to design a conditional diffusion models in the decomposed drug space which ensures flexible and fine-grained control over each substructure. We highlight our main contributions as follows:

- We propose a new molecular generation paradigm, which combines diffusion models with iterative optimization to learn molecular grammar and optimize desired properties simultaneously.
- We design a unified generation framework for *de novo* design and controllable generation via a controllable and decomposed diffusion model.
- For *de novo* design, our method can generate ligands with an average Vina Dock score of $-8.98$ and a Success Rate of $52.5\%$, achieving a new SOTA on the CrossDocked2020 benchmark.
- For controllable generation, our method shows promising results in various practical tasks of SBDD, including R-group design and scaffold hopping.

## 2 RELATED WORK

**Molecule Generation** Deep generative models have shown promising results in molecule generation. In the last decade, researchers have explored various representations and models for molecule generation. Molecules can be represented in 1D (e.g., SMILES (Weininger, 1988), SELFIES (Krenn et al., 2020)), 2D (i.e., molecular graph (Bonchev, 1991)), and 3D. Among them, 3D representations attract recent attention since they capture the complete information of molecules, and have better potential to generate and optimize molecules with regard to 3D properties, such as bioactivity for a given target (Baillif et al., 2023).

SBDD represents an important application for 3D molecule generation. Ragoza et al. (2022) generate 3D molecules in atomic density grids using a variational autoencoder (Kingma & Welling, 2013).

Luo et al. (2021); Liu et al. (2022); Peng et al. (2022) propose to generated atoms (and bonds) auto-regressively in 3D space, while Zhang et al. (2022) use fragment as building blocks instead. Guan et al. (2023a); Lin et al. (2022); Schneuing et al. (2022) introduce SE(3)-equivariant diffusion models for SBDD. More recent works have incorporated domain knowledge into 3D generative models, such as the correspondence between local fragments and subpockets. Guan et al. (2023a) suggest to break ligands into substructures and model them using decomposed priors in a diffusion framework, leading to remarkably improved binding affinities of the generated molecules. Zhang & Liu (2023) propose a subpocket prototype-augmented 3D molecule generation scheme to establish the relation between subpockets and their corresponding fragments. Existing methods based on deep generative models are powerful at distribution learning. However, when the training examples do not have the desired properties, these models can hardly generate out-of-distribution samples with these properties.

**Molecule Optimization** Optimization-based algorithms are another popular approach to design drug molecules. Methods within this category rely on predefined computable objectives to guide the optimization. Various optimization methods have been proposed for 2D drug design. JTVAE (Jin et al., 2018) uses Bayesian optimization in the latent space to indirectly optimize molecules. Reinforcement learning is used to manipulate SMILES strings (Olivecrona et al., 2017) and 2D molecular graphs (Zhou et al., 2019; Jin et al., 2020). MARS (Xie et al., 2021) leverages adaptive Markov chain Monte Carlo sampling to accelerate the exploration of chemical space. RetMol develops a retrieval-based generation scheme for iteratively improving molecular properties. Genetic algorithm is also a popular choice. GA+D (Nigam et al., 2020) uses deep learning enhanced genetic algorithm to design SELFIES strings. Graph-GA (Jensen, 2019) conducts genetic algorithm on molecular graph representation. GEGL (Ahn et al., 2020) adopts genetic algorithm to generate high quality samples for imitation learning by deep neural networks. AutoGrow 4 (Spiegel & Durrant, 2020) and RGA (Fu et al., 2022) are genetic algorithms for SBDD which incorporate target structures in molecular optimization. Both of them use molecular docking score as an objective to optimize the fitness between target structure and the generated ligands. In addition, RGA uses neural models to stablize genetic algorithm and includes target structures as a condition in its modeling. To our best knowledge, there are limited efforts on generating 3D molecules using molecular optimization.

Although optimization algorithms offers a direct approach to achieve desired properties, they require computable and accurate objectives to guide the exploration. However, not all desired properties for drug design can be easily formulated as objectives, such as molecular validity. Considering the benefits of both generative models and optimization algorithm, it is reasonable to combine them to achieve further enhanced results.

**Controllable Generation** *De novo* design aims to generate molecules from scratch, and the above-mentioned methods mainly focus on this task. Besides it, another line of research focuses on controllable molecule generation, which requires generating or optimizing partial molecules. R-group design is a task to decorate a fixed scaffold with fragments to enhance the desired properties. Langevin et al. (2020) and Maziarz et al. (2022); Imrie et al. (2021) propose to constrain the scaffold region using SMILES-based and 2D graph-based models. However, similar attempts have been rarely observed in 3D molecule generation. Scaffold hopping, on the other hand, requires the replacement of the scaffold to explore novel chemical space while maintaining the favorable decorative substructures. Imrie et al. (2020; 2021) propose autoregressive models to design 2D linker conditioning on geometric features of the input fragments and pharmacophores. Huang et al. (2022); Igashov et al. (2022) extend the application to the 3D space using variational autoencoders and diffusion models. However, existing 3D controllable generation methods are specifically designed for a single task. There lacks an unified framework to cover all possible conditional generation tasks, as well as *de novo* design.

## 3 METHOD

In this section, we will present our method, named DECOMPOPT, as illustrated in Figure 2. In Section 3.1, we show how to design a controllable and decomposed diffusion model that can generate ligand molecules conditioning on both protein subpockets and reference arms. In Section 3.2, we show how to efficiently optimize the properties of the generated ligand molecules in the decomposed drug space by improving the arm conditions.

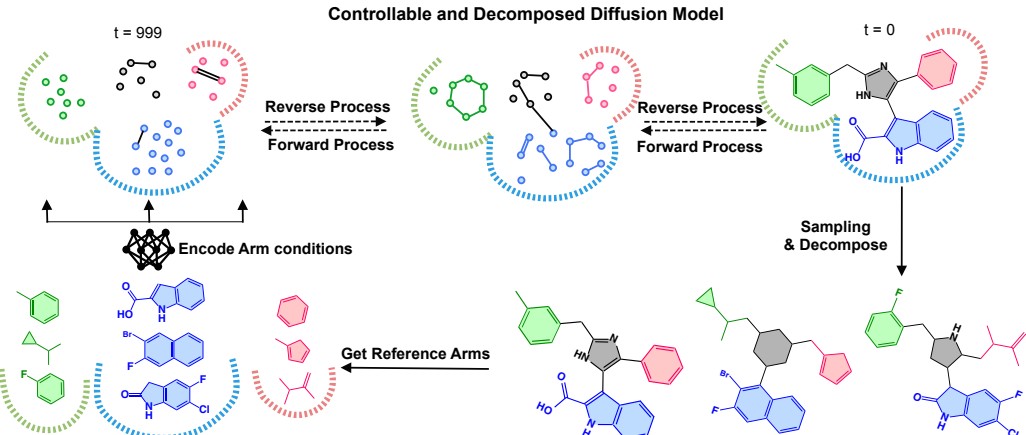

Figure 2: Illustration of DECOMPOPT. In each iteration of optimization: (1) For each subpocket, a reference arm is sampled from the ordered arm list. (2) The controllable and decomposed diffusion model generated ligand molecules based on arm (and subpocket) conditions. (3) The generated ligand molecules are collected and further decomposed into scaffolds and arms. (4) Poor arms in the ordered arm lists are replaced with the new arms that show better properties.

## 3.1 CONTROLLABLE AND DECOMPOSED DIFFUSION MODELS

A ligand molecule that binds to a specific protein pocket can be naturally decomposed into several components (i.e., arms and scaffold). The arms of a ligand molecule locally interact with subpockets of the target protein. Notably, they are the main contributors to binding affinity. The scaffold links all the arms to form a complete molecule. Inspired by this, Guan et al. (2023b) introduced decomposed priors to diffusion models for SBDD. The decomposed priors not only induce a better variational lower bound as the training objective but also provides possibilities to achieve controllability in molecular generation. Specifically, the decomposed priors allow for relatively independent modeling of each arm. To combine generative models with optimization, a flexible and controllable generation framework is need. Thus we propose a controllable and decomposed diffusion model that allows for fine-grained control over the arms of the generated ligands. Considering the different functionalities of the arms and scaffold, we only control the arms that play important roles in interaction with pockets, and leave room for the generative model on the scaffold to achieve the trade-off between controllability and diversity.

Provided with a target binding site that can be represented as $\mathcal{P} = \{(\boldsymbol{x}_i^{\mathcal{P}}, \boldsymbol{v}_i^{\mathcal{P}})\}_{i \in \{1,\ldots,N_{\mathcal{P}}\}}$, we aim to generate a ligand molecule that can be represented as $\mathcal{M} = \{(\boldsymbol{x}_i^{\mathcal{M}}, \boldsymbol{v}_i^{\mathcal{M}}, \boldsymbol{b}_{ij}^{\mathcal{M}})\}_{i,j \in \{1,\ldots,N_{\mathcal{M}}\}}$. $N_{\mathcal{P}}$ and $N_{\mathcal{M}}$ are the number of atoms in the protein pocket and ligand molecule, respectively. Here $\boldsymbol{x} \in \mathbb{R}^3$, $\boldsymbol{v} \in \mathbb{R}^d$, $\boldsymbol{b} \in \mathbb{R}^5$ denote the atom position, atom type, and bond type, respectively. A ligand molecule $\mathcal{M}$ can be decomposed into a scaffold $\mathcal{S}$ and several arms $\{\mathcal{A}_k\}_{k \in \{1,\ldots,K\}}$, where $\mathcal{M} = \mathcal{S} \cup \mathcal{A}_1 \cup \cdots \cup \mathcal{A}_K$. We denote the subpocket that is within 10Å of the atoms of the $k$-th arm $\mathcal{A}_k$ as $\mathcal{P}_k$. The controllable and decomposed diffusion model is expected to generate a ligand molecule $\mathcal{M}$ given a protein pocket $\mathcal{P}$ and several reference arms $\{\mathcal{A}_k\}$, which can be formulated as modeling the conditional distribution $q(\mathcal{M}|\mathcal{P}, \{A_k\})$. And the arms of the generated ligand molecules are expected to be similar to the corresponding reference arms.

Generally, there are two critical modules in our model: a condition encoder and a diffusion-based decoder. We employ an SE(3)-equivariant neural network (Satorras et al., 2021), named EGNN, to encode a reference arm $\mathcal{A}_k$ and its surrounding subpocket $\mathcal{P}_k$. We introduce subpockets here to include information of intermolecular interaction and relative positions. Specifically, we build a $k$-nearest-neighbor (knn) geometric graph on the complex of the reference arm and its surrounding subpocket and apply the EGNN to learn its representation as follows: $[\boldsymbol{A}_k, \boldsymbol{P}_k] = \text{Enc}(\mathcal{A}_k, \mathcal{P}_k)$, where $\boldsymbol{A}_k \in \mathbb{R}^{|\mathcal{A}_k| \times D}$, $\boldsymbol{P}_k \in \mathbb{R}^{|\mathcal{P}_k| \times D}$, $[\cdot]$ denotes concatenation along the first dimension. Each row of $\boldsymbol{A}_k$ (resp. $\boldsymbol{P}_k$) corresponds to a condition feature of an atom in the reference arm $\mathcal{A}_k$ (resp. the subpocket $\mathcal{P}_k$). $\boldsymbol{a}_k = \text{Agg}([\boldsymbol{A}_k, \boldsymbol{P}_k]) \in \mathbb{R}^D$ is the global SE(3)-invariant condition feature aggregated from the atom-wise condition features.

The diffusion model first perturbs samples by iteratively injecting random noises which are independent of the arm conditions. This leads to the forward process as follows:

$$q(\mathcal{M}_{1:T}|\mathcal{M}_0, \mathcal{P}, \{\mathcal{A}_k\}) = q(\mathcal{M}_{1:T}|\mathcal{M}_0, \mathcal{P}) = \prod_{t=1}^{T} q(\mathcal{M}_t|\mathcal{M}_{t-1}, \mathcal{P}), \tag{1}$$

where $\mathcal{M}_0 \sim q(\mathcal{M}|\mathcal{P}, \{\mathcal{A}_k\})$ and $\{\mathcal{M}_t\}_{t=1}^{T}$ is a sequence of perturbed ligands. We will introduced the aforementioned condition features into the reverse (generative) process of the diffusion model as:

$$p_\theta(\mathcal{M}_{0:T-1}|\mathcal{M}_T, \mathcal{P}, \{\mathcal{A}_k\}) = \prod_{t=1}^{T} p_\theta(\mathcal{M}_{t-1}|\mathcal{M}_t, \mathcal{P}, \{\mathcal{A}_k\}). \tag{2}$$

To model $p_\theta(\mathcal{M}_{t-1}|\mathcal{M}_t, \mathcal{P}, \{\mathcal{A}_k\})$, for the input (i.e., the ligand molecule being generated at time step $t$) of the diffusion-based decoder, we denote the SE(3)-invariant feature of its each arm atom as $\boldsymbol{v}_i^{\mathcal{A}}$ and each scaffold atom as $\boldsymbol{v}_i^{\mathcal{M}}$. For each arm atom that belongs to its $k$-th arm, we incorporate the aforementioned arm condition as $\tilde{\boldsymbol{v}}_i^{\mathcal{A}} = \text{MLP}([\boldsymbol{v}_i^{\mathcal{A}}, \boldsymbol{a}_k])$. For each scaffold atom, we do not introduce any condition (i.e., $\tilde{\boldsymbol{v}}_i^{\mathcal{S}} \coloneqq \text{MLP}(\boldsymbol{v}_i^{\mathcal{S}})$) and leave enough room for the generative model to generate diverse scaffolds. For each atom of the protein pocket, we let $\boldsymbol{v}_i^{\mathcal{P}} = \text{MLP}([\boldsymbol{v}_i^{\mathcal{P}}, \text{Agg}([\text{MLP}(\boldsymbol{v}_i^{\mathcal{P}_1}), \cdots, \text{MLP}(\boldsymbol{v}_i^{\mathcal{P}_K})])])$, where $\boldsymbol{v}_i^{\mathcal{P}_k}$ is the atom condition feature that corresponds to a specific row of $\boldsymbol{P}_k$ if this atom belongs to the $k$-th subpocket and is set to be $\boldsymbol{0}$ otherwise. For the SE(3)-equivariant feature (i.e., the coordinate in 3D space) of each atom, we do not introduce any conditions. Nevertheless, the geometric information is embedded in the SE(3)-invariant features thanks to the particular design of EGNN. After the input feature is augmented by the condition, the rest of the diffusion-based decoder mainly follows DecompDiff (Guan et al., 2023b), including the decomposed prior distribution, model architecture, training loss, etc.

To shed lights on the insights behind the dedicated model design, we will discuss more about our special considerations as follows. The encoded conditions follow the principle of decomposition. It is notable that different reference arms that locally interact with different subpockets are encoded and play roles in parallel, which allows us to control the generation more flexibly. For example, we can control the different arms of the generated ligand molecules separately. Another concern is about diversity. We do not explicitly introduce any regularization like VAE (Kingma & Welling, 2013) over the representation space of conditions in consideration of the unsmooth nature of chemical space. Nevertheless, the source of randomness is two-fold: the sampling procedure of the diffusion model and the degree of freedom of scaffold. Notably, the scaffold and arms will impact each other in the generative process and the randomness of scaffolds will also flow into arms. Expectedly, the arm of the generated ligand molecule will be similar to its corresponding reference arm but not exactly the same, which is the workhorse of our framework. This characteristic reflects both the abilities of exploration and exploitation, which is critical to the optimization process that will be introduced in the following.

---

**Algorithm 1** Optimization Process

**Input:** A specific protein pocket $\mathcal{P}$ with detected subpockets $\{\mathcal{P}_k\}_{k=1,\ldots,K}$, a reference ligand $\mathcal{M}$, pre-trained decomposed and controllable diffusion model denoted as $\text{Enc}(\cdot)$ and $\text{DiffDec}(\cdot)$

**Output:** $\text{OAL}(\mathcal{P}_k)$

 # Intialize all ordered arm lists for arms.
 $\{\mathcal{A}_k\} \leftarrow \text{Decompose}(\mathcal{M}, \mathcal{P}, \{\mathcal{P}_k\})$
 **for** $k \leftarrow 1$ to $K$ **do**
  # $s_k$ is the evaluated score.
  $(s_k, \mathcal{A}_k) \leftarrow \text{DockEval}(\mathcal{A}_k, \mathcal{P}_k)$
  $\text{OAL}(\mathcal{P}_k) \leftarrow \langle(s_k, \mathcal{A}_k)\rangle$
 **end for**
 # Start iterative optimization.
 **for** $i \leftarrow 1$ to $N$ **do**
  # The inner loop is just for better illustration
  # and is paralled as a batch in practice.
  **for** $j \leftarrow 1$ to $B$ **do**
   Sample $\{\mathcal{A}_k\}$ from $\text{OAL}(\mathcal{P}_k)$
   $\mathcal{M} \leftarrow \text{DiffDec}(\mathcal{P}, \{\text{Enc}(\mathcal{A}_k, \mathcal{P}_k)\})$
   $\{\mathcal{A}_k\} \leftarrow \text{Decompose}(\mathcal{M}, \mathcal{P}, \{\mathcal{P}_k\})$
   **for** $k \leftarrow 1$ to $K$ **do**
    $(s_k, \mathcal{A}_k) \leftarrow \text{DockEval}(\mathcal{A}_k, \mathcal{P}_k)$
    Append $(s_k, \mathcal{A}_k)$ to $\text{OAL}(\mathcal{P}_k)$
    Sort $\text{OAL}(\mathcal{P}_k)$ by $s_k$
    Keep top-$M$ elements in $\text{OAL}(\mathcal{P}_k)$
   **end for**
  **end for**
 **end for**

---

## 3.2 OPTIMIZATION IN THE DECOMPOSED DRUG SPACE

Thanks to the controllability and decomposition of the model introduced above, we can optimize the ligand molecules in the decomposed drug space. We will introduce the optimization process of DECOMPOPT as follows.

Due to the characteristics of decomposition, different arms that locally interact with different sub-pockets can be evaluated seperately. Thus we can define a score for each arm, which can be a single objective or a value that scalarizes multiple objectives by weighted sum. The following optimization process is oriented towards a given protein pocket. For each subpocket of the given protein pocket, we build an ordered list with a certain max size to store potential arms sorted by their scores. We can initialize the ordered arm lists (OAL) by decomposing reference ligands or ligand molecules generated by generative models. In each iteration of optimization, we use the controllable and decomposed diffusion models to generate a batch of ligand molecules conditioned on reference arms sampled from the ordered arm lists and further decompose them to get new arm candidates. The new arms are first refined by re-docking and then evaluated by oracles. Here we introduce an optional re-docking procedure to assign each arm with a higher-quality pose so that better interaction information can be injected into arm conditions. Then the new arms are inserted into the corresponding ordered lists and the arms with bad scores will be remove to keep predefined max size. As the process goes on, more and more arms that can well interact with the subpockets will be discovered.

The optimization process based DECOMPOPT is summarized as Algorithm 1. This optimization process shares similar ideas with well-recognized evolutionary algorithms. Arms in each ordered arm list evolve towards desired properties. The controllable generation can be viewed as one kind of mutation. But, differently, generative mutation is more efficient than the widely-used mutation defined by rule-based perturbation.

## 4 EXPERIMENTS

### 4.1 EXPERIMENTAL SETUP

**Dataset** We utilized the CrossDocked2020 dataset (Francoeur et al., 2020) to train and evaluate our model. Additionally, we adopted the same filtering and splitting strategies as the previous work (Luo et al., 2021; Peng et al., 2022; Guan et al., 2023a). The strategy focuses on retaining high-quality complexes (RMSD $< 1$Å) and diverse proteins (sequence identity $< 30\%$), leading to $100,000$ protein-binding complexes for training and $100$ novel protein for testing.

**Implementation Details** For iterative optimization, we select arms with desired properties as conditions. We initialize the list of arms with 20 molecules generated by DecompDiff in our experiment. To better align with the practical requirements of pharmaceutical practice, where the goal is to generate molecules with high drug likeness, synthesis feasibility, binding affinity, and other pertinent properties, we introduced a multi-objective optimization score to effectively balance these different objectives. In each iteration, we evaluate QED, SA, Vina Min (which will be introduced in detail later) of decomposed arms from generated molecules, then calculate Z-score$(x_i) = (x_i - \text{mean}(X))/\text{std}(X)$, $x_i \in X$ (also known as the standard score (Zill, 2020)) of each property, where $X$ denotes the set of evaluated values of a specific property. The Z-scores of each property are aggregated with same weights as criteria for selecting the top-$M$ arms as next iteration conditions. In our experiment, we conduct 30 rounds of optimization and sample 20 molecules in each round. For top-$M$ arm selection, we set $M=3$. For each protein pocket, if the average properties of sampled ligand molecules are no longer improved, the optimization process will be early stopped.

**Baselines** There are two types of baselines from the perspective of generation and optimization, respectively. **Generation Perspective**: We compare our model with various representative generative baselines: **liGAN** (Ragoza et al., 2022) is a 3D CNN-based conditional VAE model which generate ligand molecules in atomic density grids. **AR** (Luo et al., 2021), **Pocket2Mol** (Peng et al., 2022) and **GraphBP** (Liu et al., 2022) are GNN-based methods that generate 3D molecules atom by atom in an autoregressive manner. **TargetDiff** (Guan et al., 2023a) is a diffusion-based method which generates atom coordinates and atom types in a non-autoregressive way, and the prior distribution is a standard Gaussian and bonds are generated with OpenBabel (O'Boyle et al., 2011). **DecompDiff** (Guan et al., 2023b) is a diffusion-based method with decomposed priors and validity guidance which generates atoms and bonds of 3D ligand molecules in an end-to-end manner. Decompdiff has three optional decomposed priors: reference priors, pocket priors, and optimal priors. Our method also follows this setting. **Optimization Perspective**: We choose the most related work, **RGA** (Fu et al., 2022), which

Table 1: Summary of different properties of reference molecules and molecules generated by our model and other generation (Gen.) and optimization (Opt.) baselines. (↑) / (↓) denotes a larger / smaller number is better. Top 2 results are highlighted with **bold text** and underlined text, respectively.

| | Methods | Vina Score (↓) | | Vina Min (↓) | | Vina Dock (↓) | | High Affinity (↑) | | QED (↑) | | SA (↑) | | Diversity (↑) | | Success Rate (↑) |
|---|---|---|---|---|---|---|---|---|---|---|---|---|---|---|---|---|
| | | Avg. | Med. | Avg. | Med. | Avg. | Med. | Avg. | Med. | Avg. | Med. | Avg. | Med. | Avg. | Med. | |
| | Reference | -6.36 | -6.46 | -6.71 | -6.49 | -7.45 | -7.26 | - | - | 0.48 | 0.47 | 0.73 | 0.74 | - | - | 25.0% |
| Gen. | LiGAN | - | - | - | - | -6.33 | -6.20 | 21.1% | 11.1% | 0.39 | 0.39 | 0.59 | 0.57 | 0.66 | 0.67 | 3.9% |
| | GraphBP | - | - | - | - | -4.80 | -4.70 | 14.2% | 6.7% | 0.43 | 0.45 | 0.49 | 0.48 | **0.79** | **0.78** | 0.1% |
| | AR | -5.75 | -5.64 | -6.18 | -5.88 | -6.75 | -6.62 | 37.9% | 31.0% | 0.51 | 0.50 | 0.63 | 0.63 | 0.70 | 0.70 | 7.1% |
| | Pocket2Mol | -5.14 | -4.70 | -6.42 | -5.82 | -7.15 | -6.79 | 48.4% | 51.0% | 0.56 | 0.57 | **0.74** | **0.75** | 0.69 | 0.71 | 24.4% |
| | TargetDiff | -5.47 | -6.30 | -6.64 | -6.83 | -7.80 | -7.91 | 58.1% | 59.1% | 0.48 | 0.48 | 0.58 | 0.58 | 0.72 | 0.71 | 10.5% |
| | DecompDiff | -5.67 | -6.04 | -7.04 | -7.09 | -8.39 | -8.43 | 64.4% | 71.0% | 0.45 | 0.43 | 0.61 | 0.60 | 0.68 | 0.68 | 24.5% |
| Opt. | RGA | - | - | - | - | -8.01 | -8.17 | 64.4% | 89.3% | 0.57 | 0.57 | 0.71 | 0.73 | 0.41 | 0.41 | 46.2% |
| Gen. + Opt. | TargetDiff w/ Opt. | **-7.87** | **-7.48** | **-7.82** | -7.48 | -8.30 | -8.15 | 71.5% | 95.9% | **0.66** | **0.68** | 0.68 | 0.67 | 0.31 | 0.30 | 25.8% |
| | DECOMPOPT | -5.87 | -6.81 | -7.35 | **-7.72** | **-8.98** | **-9.01** | 73.5% | 93.3% | 0.48 | 0.45 | 0.65 | 0.65 | 0.60 | 0.61 | **52.5%** |

is a reinforced genetic algorithm with policy networks and also focus on structure-based molecular optimization, as the baseline. Besides, we also introduced the controllability into TargetDiff by whole molecule conditions rather than arm conditions. We name this baseline as **TargetDiff w/ Opt.**.

**Metircs** We evaluate the generated molecules from **target binding affinity and molecular properties**. We employ AutoDock Vina (Eberhardt et al., 2021) to estimate the target binding affinity, following the same setup as Luo et al. (2021); Ragoza et al. (2022). We collect all generated molecules across 100 test proteins and report the mean and median of affinity-related metrics (*Vina Score*, *Vina Min*, *Vina Dock*, and *High Affinity*) and property-related metrics (drug-likeness *QED* (Bickerton et al., 2012), synthesizability *SA* (Ertl & Schuffenhauer, 2009), and *diversity*). Vina Score directly estimates the binding affinity based on the generated 3D molecules, Vina Min conducts a local structure minimization before estimation, Vina Dock involves a re-docking process and reflects the best possible binding affinity, and High Affinity measures the percentage of how many generated molecules binds better than the reference molecule per test protein. Following Yang et al. (2021); Long et al. (2022); Guan et al. (2023b); Yang et al. (2024), we further report the percentage of molecules which pass certain criteria (QED > 0.25, SA > 0.59, Vina Dock < −8.18) as *Success Rate* for comprehensive evaluation, considering that practical drug design also requires the generated molecules to be drug-like, synthesizable, and maintain high binding affinity simultaneously (Jin et al., 2020; Xie et al., 2021). Additionally, we also evaluate the generated molecules from the perspective of molecular conformation. Please refer to Appendix B for more results.

## 4.2 MAIN RESULTS

We evaluate the effectiveness of our model in terms of binding affinity and molecular properties. As shown in Table 1, DECOMPOPT outperforms baselines by a large margin in affinity-related metrics and Success Rate, a comprehensive metric. Specifically, DECOMPOPT surpasses the strong baseline DecompDiff in all metrics, except diversity. All these gains clearly indicate that the optimization works as we expected and our method better aligns with the goal of drug discovery compared with the generative baselines.

Notably, RGA, which also focuses on structure-based molecular optimization, achieves promising Sucess Rate. And our method performs even better. More importantly, molecular optimization methods like RGA inevitably encounter the pitfalls of low diversity and inefficiency. Because these optimization methods usually start with a reference ligand and iteratively make small changes on it to gradually improve the property. They are easy to be trapped in local solutions. However, DECOMPOPT inherites advantages of both optimization and generation, achieving high Success Rate and considerable diversity at the same time. See Appendix C for additional results.

## 4.3 ABLATION STUDIES

**Single-Objective Optimization** To further validate the effectiveness of DECOMPOPT, we test our method on the setting of single-objective optimization with reference priors. Specifically, we use QED, SA, and Vina Min as the objective and analyze the results of three experiments, respectively. As shown in Table 2, each experiment effectively improves the corresponding property.

**Benefits of Decomposed Optimization** The decomposed drug space allows for optimizing each arm of a ligand molecule separately. In our method, arms of each subpockets are evaluated and

Table 2: Summary of results of single-objective optimization. The improvements of DECOMPOPT over the baseline are highlighted in green.

| Property | DecompDiff | | DECOMPOPT | |
|---|---|---|---|---|
| | Avg. | Med. | Avg. | Med. |
| QED (↑) | 0.48 | 0.48 | 0.52 (+8.3%) | 0.53 (+10.4%) |
| SA (↑) | 0.67 | 0.66 | 0.74 (+10.5%) | 0.74 (+12.1%) |
| Vina Min (↓) | -6.01 | -5.98 | -6.72 (+11.8%) | -6.72 (+12.4%) |

Table 3: Comparison of optimization strategies using molecule-level and arm-level conditions.

| Method | Vina Min (↓) | |
|---|---|---|
| | Avg. | Med. |
| DecompDiff | -6.01 | -5.98 |
| Molecule-level Opt. | -6.62 | -6.66 |
| Arm-level Opt. | -6.72 | -6.72 |

selected independently (namely, arm-level optimization). We also tried evaluating the property of the whole ligand molecules, choosing those with desired property, and decomposing them to serve as the arm conditions in the next optimization iteration (namely, molecule-level optimization). We compare these two optimization under the setting of reference prior. As shown in Table 3, arm-wise optimization performs better than molecule-wise optimization, which demonstrates benefits brought by decomposition in terms of optimization efficiency.

## 4.4 CONTROLLABILITY

Various molecular optimization scenarios, including R-group design and scaffold hopping, play a crucial role in real-world drug discovery. They enhance binding affinity, potency, and other relevant molecular properties with greater precision. Our controllable and decomposed diffusion model seamlessly integrates with these scenarios by incorporating expert knowledge through decomposed arm conditions, better aligned with the demands and objectives of the pharmaceutical industry.

Table 4: Scaffold hopping results of Decompdiff and DECOMPOPT on CrossDocked2020 test set.

| Methods | Valid (↑) | Unique (↑) | Novel (↑) | Complete Rate (↑) | Scaffold Similarity (↓) |
|---|---|---|---|---|---|
| DecompDiff+Inpainting | 0.95 | 0.48 | 0.85 | 89.2% | 0.40 |
| DECOMPOPT +Inpainting | 0.96 | 0.46 | 0.88 | 93.0% | 0.35 |

**R-group Design**   R-group optimization is a widely used technique to optimize molecules' substituents for improving biological activity. Our model is well-suited for this task by employing finer-level arm conditions to guide the optimization. To achieve the optimization goal, we decompose the compound into a scaffold and multiple arms. Subsequently, we choose a specific arm for optimization to enhance its binding affinity. The optimization process involves conditional inpainting (Lugmayr et al., 2022) inspired by Schneuing et al. (2022).

We diffuse the remaining part of the compound while predicting the selected arm, which is conditioned on the reference arm at each step. We initialize the list of arms by the arms defined as r-group to optimize. From the molecules generated through this process, we can select substituents with higher Vina Min Score to serve as the condition for the next iteration. Results of R-group optimization on protein 3DAF and 4F1M are presented in Figure 3. More results on protein 4G3D can be found in Appendix D. After 30 rounds of optimization, our generated molecules achieve a vina minimize score more than 1 kcal/mol better than the reference. Moreover, we compare DECOMPOPT with Decompdiff for R-group optimization on Vina minimize, Tanimoto similarity, and Complete rate with detailed result in Appendix D. Tanimoto similarity is calculated using rdkit `GetMorganFingerprint` and `TanimotoSimilarity` functions. Complete rate measures the proportion of completed molecules in generated result. As Table 14 shows, the decomposed conditional arms bring greater controllability over the shape, positioning, and properties of the generated substituents compared to diffusion inpainting.

Fragment growing is a another technique for R-group design. Unlike R-group optimization, fragment growing aims to design new substituents instead of optimization existing ones. By designing novel arms priors and predicting the number of atoms through chemistry software or expert guidance, DECOMPOPT can naturally facilitate incremental growth and optimization of newly generated arms, leading to improved biological activity. A case study on the application of fragment growing can be found in Appendix D.

**Scaffold Hopping** In contrast to R-group design, scaffold hopping involves the generation of scaffolds to connect pivotal groups, which play a critical role in interaction with proteins. We apply our decomposed conditional diffusion model to scaffold hopping by inpainting scaffold in fixed arms and incorporating pivotal arms as structural conditions. We generate 20 molecules for each target on our test set. Following Huang et al. (2022), we measure the *Validity*, *Uniqueness*, and *Novelty* of generated molecules. Additionally, we compute the *Complete Rate*, which measures the proportion of successfully constructed molecules with all atoms connected. To better understand the influence of conditional arms on scaffold generation, we estimate *Scaffold Similarity* between generated scaffold and reference scaffold following Polykovskiy et al. (2020). A detailed description of scaffold hopping evaluation metrics can be found in Appendix D. The supplementary information provided indicates the inclusion of arm conditions can influence scaffold generation through message passing, leading to a more consistent scaffold when compared to diffusion inpainting without arm conditions. As Table 4 shows, our model achieves higher validity and complete rate. Higher novelty and lower scaffold similarity indicate that our model is better at maintaining diversity and exploring novel molecules while controlling pivotal groups. In Figure 4, we show results of scaffold hopping on 1R1H and 5AEH.

More visualization results can be found in Appendix D.

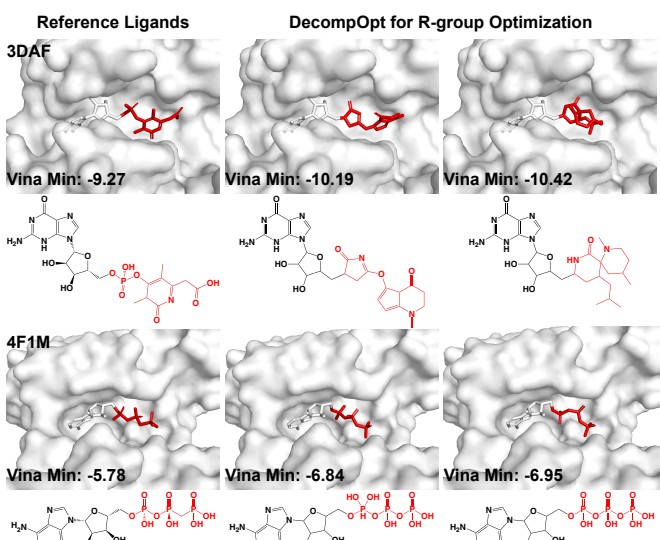

Figure 3: Visualization of reference binding molecules (left column), molecules generated by DECOMPOPT (middle and right column) with 30 rounds of optimization on protein 3DAF (top row) and 4F1M (bottom row). Optimized R-group are highlighted in red.

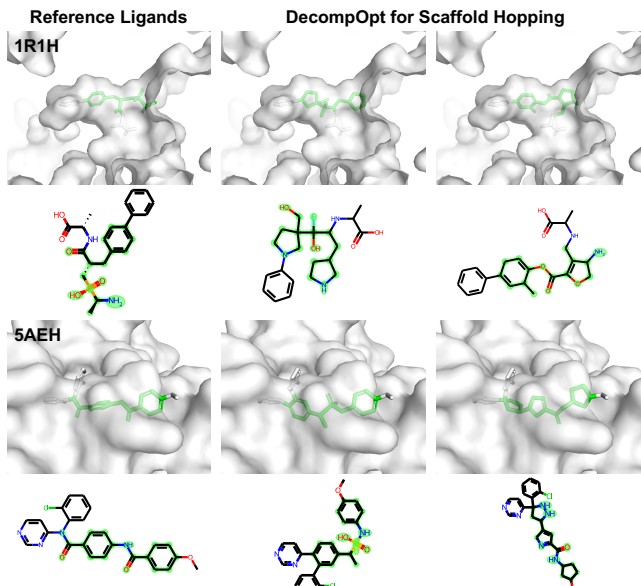

Figure 4: Examples of scaffold hopping accomplished by DE-COMPOPT. For each row, the left image shows the reference ligand, the middle and right images are two examples generated by DECOMPOPT. Reference and generated scaffolds are highlighted in green.

## 5 CONCLUSIONS

In this work, we proposed a controllable and decomposed diffusion model for structure-based molecular optimization and opened a new paradigm that combines generation and optimization for structure-based drug design. Our method shows promising performance on both *de novo* design and controllable generation, indicating its great potential in drug discovery. We would like to point out that in our current controllable diffusion models, we did not explore the best way for multi-objective optimization. We plan to investigate it in our future work.

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

# A IMPLEMENTATION DETAILS

In this section, we will provide more implementation details of our methods. Though some contents, such as the SE(3)-equivariant layer and the loss function, can be referred to Guan et al. (2023b), we still include them here to make our paper more self-containing.

## A.1 FEATURIZATION

We follow the decomposition algorithm proposed by Guan et al. (2023b) to decompose ligand molecules into arms and a scaffold. We define the part of proteins that lies within 10Å of any atom of an arm as its corresponding subpocket.

Following DecompDiff (Guan et al., 2023b), we represent each protein atom with the following features: one-hot element indicator (H, C, N, O, S, Se), one-hot amino acid type indicator (20 dimension), one-dim flag indicating whether the atom is a backbone atom, and one-hot arm/scaffold region indicator. If the distance between the protein atom and any arm center is within 10Å, the protein atom will be labeled as belonging to an arm region and otherwise a scaffold region. The ligand atom is represented with following features: one-hot element indicator (C, N, O, F, P, S, Cl) and one-hot arm/scaffold indicator. Different from DecompDiff, the atom features are enhanced by concatenating an SE(3)-invariant feature of arms and their corresponding subpockets encoded by the condition encoder after the orignal features.

Two graphs are constructed for message passing in the protein-ligand complex: a k-nearest neighbors graph $\mathcal{G}_K$ upon ligand atoms and protein atoms (we choose $k = 32$ in all experiments) and a fully-connected graph $\mathcal{G}_L$ upon ligand atoms. The edge features are the outer products of distance embedding and edge type. The distance embedding is obtained by expanding distance with radial basis functions (RBF) located at 20 centers between 0Å and 10Å. The edge type is a 4-dim one-hot vector indicating the edge is between ligand atoms, protein atoms, ligand-protein atoms or protein-ligand atoms. In the ligand graph, the ligand bond is represented with a one-hot bond type vector (non-bond, single, double, triple, aromatic), an additional feature indicating whether or not two ligand atoms are from the same arm/scaffold.

## A.2 MODEL DETAILS

The controllable and decomposed diffusion model consist of two parts: a condition encoder and a diffusion-based decoder. The building block is an SE(3)-equivariant layer that is composed of three layers: atom update layer, bond update layer, and position update layer.

We denote the protein pocket as $\mathcal{P} = \{(\mathbf{x}_i^{\mathcal{P}}, \mathbf{v}_i^{\mathcal{P}})\}_{i \in \{1,...,N_{\mathcal{P}}\}}$ and the ligand molecule as $\mathcal{M} = \{(\boldsymbol{x}_i, \boldsymbol{v}_i, \boldsymbol{b}_{ij})\}_{i,j \in \{1,...,N_{\mathcal{M}}\}}$, where $\boldsymbol{x}$ is the atom position, $\boldsymbol{v}$ is the atom type, and $\boldsymbol{b}_{ij}$ is the chemical bond type between the atom $i$ and the atom $j$. For brevity, we omit the superscript $\mathcal{P}$ or $\mathcal{M}$ in the following. We use $\mathbf{h}_i$ to denote the the SE(3)-invariant hidden state of $i$-th atom, $\mathbf{x}_i$ to denote the $i$-th atom's coordinate, which is SE(3)-equivariant, and $\mathbf{e}_{ij}$ to denote the hidden state of the edge between the $i$-th atom and the $j$-th atom. They can be obtained as we described in the previous subsection. And we use $t$ to denote the time embedding as that in Ho et al. (2020).

**Atom Update Layer** We denote the atom update layer as $\phi_a := \{\phi_{a1}, \phi_{a2}, \phi_{a3}, \phi_{a4}\}$.

We first use the atom update layer $\phi_{a1}$ to model protein-ligand interation as follows:

$$\Delta\mathbf{h}_{K,i} \leftarrow \sum_{j \in \mathcal{N}_K(i)} \phi_{a1}(\mathbf{h}_i, \mathbf{h}_j, ||\mathbf{x}_i - \mathbf{x}_j||, \mathbf{e}_{ij}, t), \tag{3}$$

where $\mathcal{N}_K(i)$ is the set of neighbors of the $i$-th atom in the protein-ligand complex graph $\mathcal{G}_K$.

We then further use the atom update layer $\phi_{a2}$ and $\phi_{a3}$ to model the interaction inside the ligand as follows:

$$\mathbf{m}_{ij} \leftarrow \phi_{a2}(||\mathbf{x}_i - \mathbf{x}_j||, \mathbf{e}_{ij}), \tag{4}$$

$$\Delta\mathbf{h}_{L,i} \leftarrow \sum_{j \in \mathcal{N}_L(i)} \phi_{a3}(\mathbf{h}_i, \mathbf{h}_j, \mathbf{m}_{ji}, t), \tag{5}$$

where $\mathcal{N}_L(i)$ represents the set of neighbors of the $i$-th atom in the ligand graph $\mathcal{G}_L$. Finally, we update the hidden state of atoms by the atom update layer $\phi_{a4}$ as follows:

$$\mathbf{h}_i \leftarrow \mathbf{h}_i + \phi_{a4}(\Delta \mathbf{h}_{K,i} + \Delta \mathbf{h}_{L,i}). \tag{6}$$

**Bond Update Layer**    We update the hidden states of the edges by the bond update layer $\phi_b$ as follows:

$$\mathbf{e}_{ij} \leftarrow \sum_{k \in \mathcal{N}_L(i) \backslash \{j\}} \phi_b(\mathbf{h}_i, \mathbf{h}_j, \mathbf{h}_k, \mathbf{m}_{kj}, \mathbf{m}_{ji}, t). \tag{7}$$

**Position Update Layer**    The atom positions are updated by the position update layer $\phi_p :=$ $\{\phi_{p1}, \phi_{p2}\}$ as follows:

$$\Delta \mathbf{x}_{K,i} \leftarrow \sum_{j \in \mathcal{N}_K(i)} (\mathbf{x}_j - \mathbf{x}_i) \phi_{p1}(\mathbf{h}_i, \mathbf{h}_j, ||\mathbf{x}_i - \mathbf{x}_j||, t), \tag{8}$$

$$\Delta \mathbf{x}_{L,i} \leftarrow \sum_{j \in \mathcal{N}_L(i)} (\mathbf{x}_j - \mathbf{x}_i) \phi_{p2}(\mathbf{h}_i, \mathbf{h}_j, ||\mathbf{x}_i - \mathbf{x}_j||, \mathbf{m}_{ji}, t), \tag{9}$$

$$\mathbf{x}_i \leftarrow \mathbf{x}_i + (\Delta \mathbf{x}_{K,i} + \Delta \mathbf{x}_{L,i}) \cdot \mathbb{1}_{\text{mol}}, \tag{10}$$

where $\mathbb{1}_{\text{mol}}$ is the indicator of ligand atoms since we assume the protein atoms are fixed as the context.

In practice, the condition encoder consists of two SE(3)-equivariant layers and the diffusion-based decoder consists of six SE(3)-equivaraint layers. In each SE(3)-equivariant layer, following Guan et al. (2023b), we apply graph attention to aggregate the message of each node/edge. The key/value/query embedding is obtained through a 2-layer MLP with LayerNorm and ReLU activation. Stacking these three layers as a block, our model consists of 6 blocks with `hidden_dim=128` and `n_heads=16`. Additionally, the diffusion-based decoder also have two prediction heads (which are simply 2-layer MLPs and following Softmax function) that maps the learned hidden states of atoms and edges to the predicted atom type and bond type.

## A.3    TRAINING DETAILS

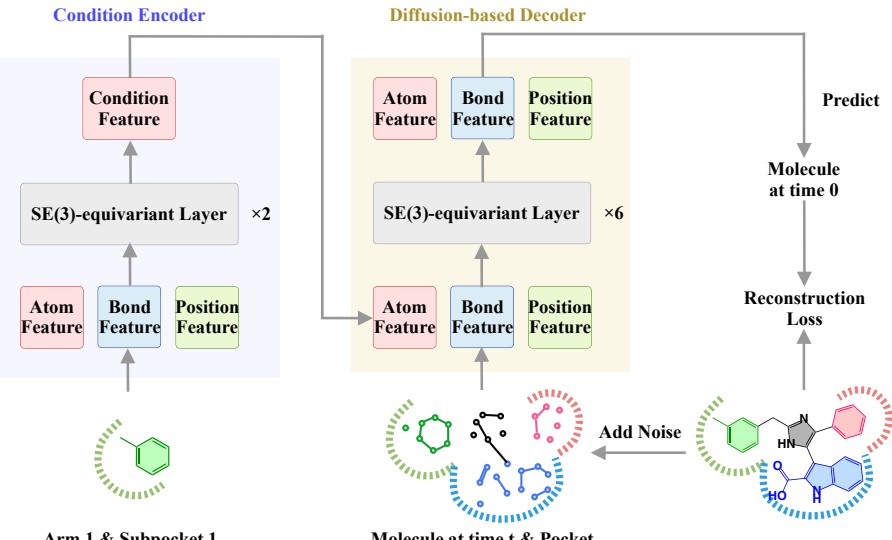

Figure 5: Illustration of training. For this case, there are actually three pairs of arms and subpockets input to the condition encoders separately. For brevity, we only plot one as an example.

Given a pair of protein and ligand molecule, we first decompose the molecule to get the arms. We add noise to the ligand molecules in the training set to get the perturbed molecules as the forward process of diffusion models (equation 1). The forward process is a Markov chain with fixed variance schedule $\{\beta_t\}_{t=1,\ldots,T}$ (Ho et al., 2020). We denote $\alpha_t = 1 - \beta_t$ and $\bar{\alpha}_t = \prod_{s=1}^{t} \alpha_s$. More specifically, the

noises at time $t$ are injected as follows:

$$q(\mathbf{x}_t|\mathbf{x}_0) = \mathcal{N}(\mathbf{x}_t; \sqrt{\bar{\alpha}_t}\mathbf{x}_0, (1 - \bar{\alpha}_t)\mathbf{I}), \tag{11}$$

$$q(\mathbf{v}_t|\mathbf{v}_0) = \mathcal{C}(\mathbf{v}_t|\bar{\alpha}_t\mathbf{v}_0 + (1 - \bar{\alpha}_t)/K_a), \tag{12}$$

$$q(\mathbf{b}_t|\mathbf{b}_0) = \mathcal{C}(\mathbf{v}_t|\bar{\alpha}_t\mathbf{b}_0 + (1 - \bar{\alpha}_t)/K_b), \tag{13}$$

where $K_a$ and $K_b$ are the number of atom classes and bond classes respectively.

Then the arms and subpockets are input to the condition encoder. The output of condition encoder and the perturbed ligand are further input to the diffusion-based decoder. Then the reconstruction loss $L_t$ at time $t$ is defined as follows:

$$L_t^{(v)} = \sum_{k=1}^{K_a} \mathbf{c}(\mathbf{v}_t, \mathbf{v}_0)_k \log \frac{\mathbf{c}(\mathbf{v}_t, \mathbf{v}_0)_k}{\mathbf{c}(\mathbf{v}_t, \hat{\mathbf{v}}_0)_k}, \tag{14}$$

$$L_t^{(b)} = \sum_{k=1}^{K_b} \mathbf{c}(\mathbf{b}_t, \mathbf{b}_0)_k \log \frac{\mathbf{c}(\mathbf{b}_t, \mathbf{b}_0)_k}{\mathbf{c}(\mathbf{b}_t, \hat{\mathbf{b}}_0)_k}, \tag{15}$$

$$L_t^{(x)} = ||\mathbf{x}_0 - \hat{\mathbf{x}}_0||^2, \tag{16}$$

$$L_t = L_t^{(x)} + \gamma_v L_t^{(v)} + \gamma_b L_t^{(b)}, \tag{17}$$

where $(\mathbf{x}_t, \mathbf{v}_t, \mathbf{b}_t)$, $(\mathbf{x}_0, \mathbf{v}_t, \mathbf{b}_t)$, and $(\hat{\mathbf{x}}_0, \hat{\mathbf{v}}_0, \hat{\mathbf{b}}_0)$ represents atom positions, atom types, and bond types of the perturbed molecule at time $t$, ground truth molecule, and the predicted molecule respectively, $\mathbf{c}(\mathbf{v}_t, \mathbf{v}_0) = \mathbf{c}^\star / \sum_{k=1}^{K_a} c_k^\star$ and $\mathbf{c}^\star(\mathbf{v}_t, \mathbf{v}_0) = [\alpha_t \mathbf{v}_t + (1 - \alpha_t)/K_a] \odot [\bar{\alpha}_{t-1}\mathbf{v}_0 + (1 - \bar{\alpha}_{t-1})/K_a]$, $\mathbf{c}(\mathbf{b}_t, \mathbf{b}_0) = \mathbf{c}^\star / \sum_{k=1}^{K_b} c_k^\star$ and $\mathbf{c}^\star(\mathbf{b}_t, \mathbf{b}_0) = [\alpha_t \mathbf{b}_t + (1 - \alpha_t)/K_b] \odot [\bar{\alpha}_{t-1}\mathbf{b}_0 + (1 - \bar{\alpha}_{t-1})/K_b]$. Note that the condition encoder and the diffusion-based decoder are jointly trained.

In practice, we set the loss weights as $\gamma_v = 100$ and $\gamma_b = 100$. Follwing the setting of Guan et al. (2023b), we set the number of diffusion steps as 1000. For this diffusion noise schedule, we choose to use a sigmoid $\beta$ schedule with $\beta_1 = \texttt{1e-7}$ and $\beta_T = \texttt{2e-3}$ for atom coordinates, and a cosine $\beta$ schedule suggested in Nichol & Dhariwal (2021) with $s = 0.01$ for atom types and bond types.

We use Adam Kingma & Ba (2014) with `init_learning_rate=0.0005`, `betas=(0.95, 0.999)` to train the model. And we set `batch_size=16` and `clip_gradient_norm=8`. During the training phase, we add a small Gaussian noise with a standard deviation of 0.1 to protein atom coordinates as data augmentation. We also schedule to decay the learning rate exponentially with a factor of 0.6 and a minimum learning rate of 1e-6. The learning rate is decayed if there is no improvement for the validation loss in 10 consecutive evaluations. The evaluation is performed for every 1000 training steps. We trained our model on one NVIDIA GeForce GTX A100 GPU, and it could converge within 237k steps.

## A.4 SAMPLING DETAILS

To sample molecules using the pre-trained controllable and decomposed diffusion model, assume that there are available arms as conditions, we can first sample a noisy molecule from the prior distribution and derive a molecule by iteratively denoising following the reverse process (equation 2). More specifically, the denoising step at time $t$ corresponds to sampling molecules from the following distributions:

$$q(\mathbf{x}_{t-1}|\mathbf{x}_t, \hat{\mathbf{x}}_0) = \mathcal{N}(\mathbf{x}_{t-1}; \tilde{\boldsymbol{\mu}}_t(\mathbf{x}_t, \hat{\mathbf{x}}_0), \tilde{\beta}_t\mathbf{I}), \tag{18}$$

$$q(\mathbf{v}_{t-1}|\mathbf{v}_t, \hat{\mathbf{v}}_0) = \mathcal{C}(\mathbf{v}_{t-1}|\tilde{\mathbf{c}}_t(\mathbf{v}_t, \hat{\mathbf{v}}_0)), \tag{19}$$

$$q(\mathbf{b}_{t-1}|\mathbf{b}_t, \hat{\mathbf{b}}_0) = \mathcal{C}(\mathbf{b}_{t-1}|\tilde{\mathbf{c}}_t(\mathbf{b}_t, \hat{\mathbf{b}}_0)), \tag{20}$$

where $\tilde{\boldsymbol{\mu}}_t(\mathbf{x}_t, \hat{\mathbf{x}}_0) = \frac{\sqrt{\bar{\alpha}_{t-1}}\beta_t}{1-\bar{\alpha}_t}\hat{\mathbf{x}}_0 + \frac{\sqrt{\alpha_t}(1-\bar{\alpha}_{t-1})}{1-\bar{\alpha}_t}\mathbf{x}_t$, $\tilde{\beta}_t = \frac{1-\bar{\alpha}_{t-1}}{1-\bar{\alpha}_t}\beta_t$, $\tilde{\mathbf{c}}(\mathbf{v}_t, \hat{\mathbf{v}}_0) = \tilde{\mathbf{c}}^\star / \sum_{k=1}^{K_a} \tilde{c}_k^\star$ and $\tilde{\mathbf{c}}^\star(\mathbf{v}_t, \hat{\mathbf{v}}_0) = [\alpha_t \mathbf{v}_t + (1 - \alpha_t)/K_a] \odot [\bar{\alpha}_{t-1}\hat{\mathbf{v}}_0 + (1 - \bar{\alpha}_{t-1})/K_a]$, $\tilde{\mathbf{c}}(\mathbf{b}_t, \hat{\mathbf{b}}_0) = \tilde{\mathbf{c}}^\star / \sum_{k=1}^{K_b} \tilde{c}_k^\star$ and $\tilde{\mathbf{c}}^\star(\mathbf{b}_t, \hat{\mathbf{b}}_0) = [\alpha_t \mathbf{b}_t + (1 - \alpha_t)/K_b] \odot [\bar{\alpha}_{t-1}\hat{\mathbf{b}}_0 + (1 - \bar{\alpha}_{t-1})/K_b]$. Here $(\hat{\mathbf{x}}_0, \hat{\mathbf{v}}_0, \hat{\mathbf{b}}_0)$ is the molecule output by the diffusion-based decoder, whose input is the noisy molecule at time $t$ and the condition feature. The sampling step is illustrated as Figure 6. During sampling, we also apply validity guidance

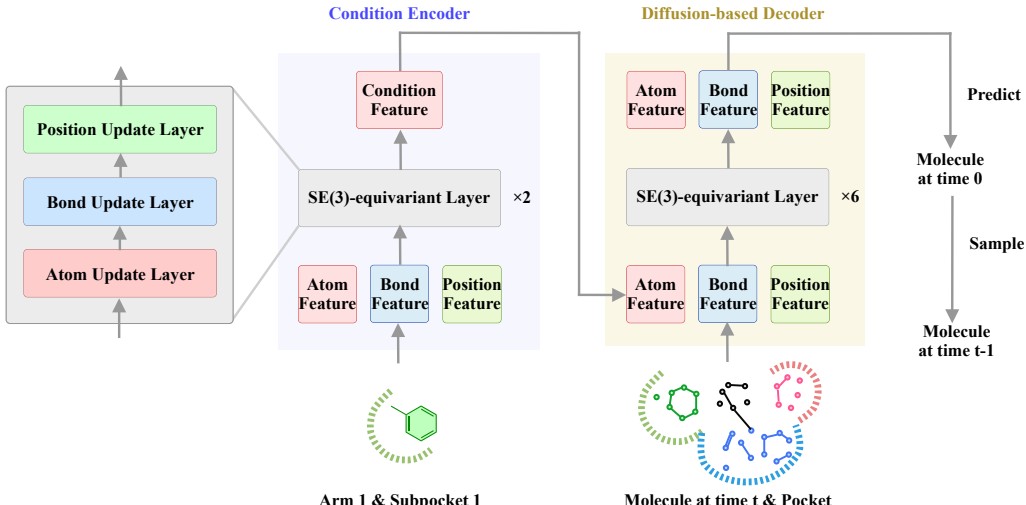

Figure 6: Illustration of the sampling.

proposed by Guan et al. (2023b), which encourages the model to generate molecules with valid structures.

## A.5 OPTIMIZATION DETAILS

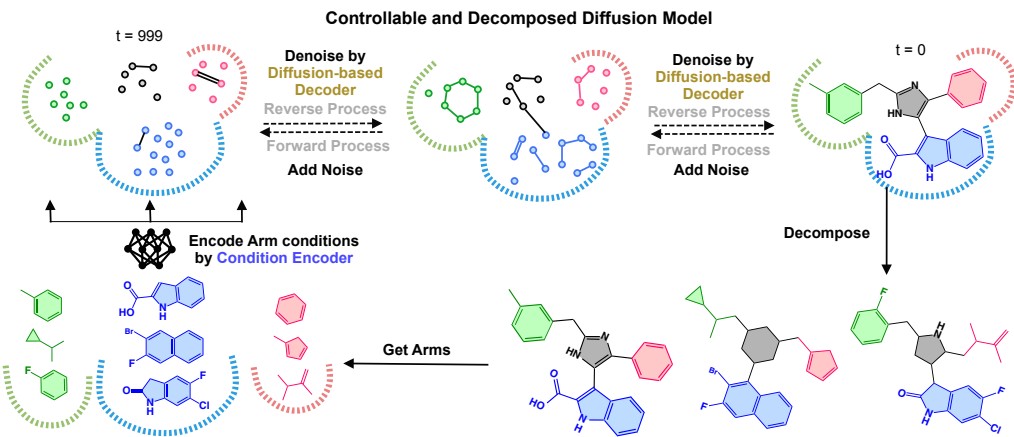

Figure 7: Illustration of molecular optimization (revised based on Figure 2). It is highlighted where we apply the condition encoder and the diffusion-based decoder.

To generate molecules with desired properties, we can apply the pre-trained controllable and decomposed diffusion models for structure-based molecular optimization without any fine-tuning. The optimization procedure is summarized as Algorithm 1 and illustrated as Figure 7. In practice, since reference ligands are not available, we can initialize the arm lists with 20 ligands generated by DecompDiff, and this is the actual setting in our experiment. We have provided the optimization procedure in detail that can be found in Section 3.2 and Section 4.1.

## B EVALUATION OF MOLECULAR CONFORMATION

To evaluate generated molecules from the perspective of molecular conformation, we compute the Jensen-Shannon divergences (JSD) in atom distance distributions between the reference molecules and the generated molecules (see Figure 8).

We also compute different bond distance and bond angle distributions of the generated molecules and compare them against the corresponding reference empirical distributions in Tables 5 and 6.

To further measure the quality of generated conformation, we optimize the generated structures with Merck Molecular Force Field (MMFF) (Halgren, 1996) and calculate the energy difference between

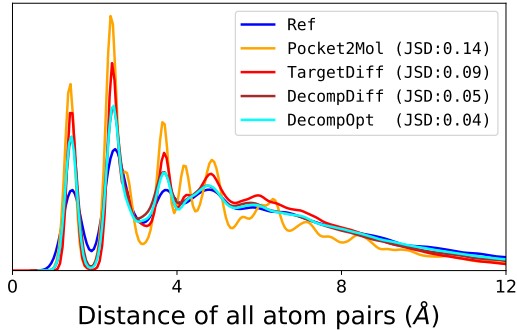

Figure 8: Comparing the distribution for distances of all-atom for reference molecules in the test set and model-generated molecules. Jensen-Shannon divergence (JSD) between two distributions is reported.

Table 5: Jensen-Shannon divergence between bond distance distributions of the reference molecules and the generated molecules, and lower values indicate better performances. "-", "=", and ":" represent single, double, and aromatic bonds, respectively. We highlight the best two results with **bold text** and underlined text, respectively.

| Bond | liGAN | GraphBP | AR | Pocket2 Mol | Target Diff | Decomp Diff | Ours |
|------|-------|---------|-----|-------|--------|--------|------|
| C−C | 0.601 | 0.368 | 0.609 | 0.496 | 0.369 | **0.359** | 0.362 |
| C=C | 0.665 | 0.530 | 0.620 | 0.561 | 0.505 | 0.537 | **0.504** |
| C−N | 0.634 | 0.456 | 0.474 | 0.416 | 0.363 | 0.344 | **0.328** |
| C=N | 0.749 | 0.693 | 0.635 | 0.629 | 0.550 | 0.584 | **0.566** |
| C−O | 0.656 | 0.467 | 0.492 | 0.454 | 0.421 | 0.376 | **0.373** |
| C=O | 0.661 | 0.471 | 0.558 | 0.516 | 0.461 | 0.374 | **0.329** |
| C:C | 0.497 | 0.407 | 0.451 | 0.416 | 0.263 | 0.251 | **0.196** |
| C:N | 0.638 | 0.689 | 0.552 | 0.487 | 0.235 | 0.269 | **0.219** |

pre- and pos- MMFF-optimized coordinates for different rigid fragments that do not contain any rotatable bonds. As Table 7 and Figure 9 show, DECOMPOPT achieves low energy differences and outperforms baselines in most cases. We also calculate the energy difference before and after force field optimization for the whole molecules. As Table 8 and Figure 10 show, notably, DECOMPOPT outperforms all diffusion-based methods by a large margin and achieve comparable performance with the best baseline. These results show that the conformation of ligands generated by DECOMPOPT is high-quality and stable.

Table 7: Median energy difference for rigid fragment of different fragment size (3/4/5/6/7/8 atoms) before and after the force-field optimization.

| Methods | Median Energy Difference (↓) | | | | | |
|---------|------|------|------|------|------|------|
|  | 3 | 4 | 5 | 6 | 7 | 8 |
| LiGAN | 86.32 | 165.15 | 105.96 | 185.70 | 243.79 | 332.81 |
| AR | 25.79 | 73.06 | 23.89 | 30.42 | 56.47 | 76.50 |
| Pocket2Mol | 10.43 | 33.93 | 34.47 | 27.86 | 33.90 | 42.97 |
| TargetDiff | 7.31 | 30.57 | 18.01 | 11.98 | 28.92 | 50.42 |
| DecompDiff | 6.01 | 29.20 | 10.78 | 4.33 | 12.74 | 30.68 |
| DECOMPOPT | 6.00 | 16.59 | 9.89 | 2.61 | 13.29 | 31.49 |

Table 6: Jensen-Shannon divergence between bond angle distributions of the reference molecules and the generated molecules, and lower values indicate better performances. We highlight the best two results with **bold text** and underlined text, respectively.

| Angle | liGAN | GraphBP | AR | Pocket2 Mol | Target Diff | Decomp Diff | Ours |
|-------|-------|---------|-----|-------------|-------------|-------------|------|
| CCC | 0.598 | 0.424 | 0.340 | 0.323 | 0.328 | 0.314 | **0.280** |
| CCO | 0.637 | 0.354 | 0.442 | 0.401 | 0.385 | 0.324 | **0.331** |
| CNC | 0.604 | 0.469 | 0.419 | **0.237** | 0.367 | 0.297 | 0.280 |
| OPO | 0.512 | 0.684 | 0.367 | 0.274 | 0.303 | 0.217 | **0.198** |
| NCC | 0.621 | 0.372 | 0.392 | 0.351 | 0.354 | 0.294 | **0.266** |
| CC=O | 0.636 | 0.377 | 0.476 | 0.353 | 0.356 | 0.259 | **0.257** |
| COC | 0.606 | 0.482 | 0.459 | **0.317** | 0.389 | 0.339 | 0.338 |

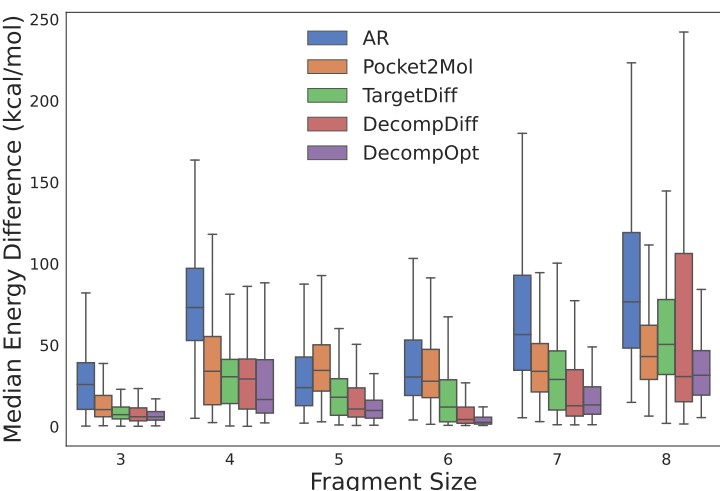

Figure 9: Median energy difference for molecules with different number of rotatable bonds before and after the force-field optimization.

Table 8: Median energy difference for molecules with different number of rotatable bonds (1/2/3/4/5/6/7 rotatable bonds) before and after the force-field optimization.

| Methods | Median Energy Difference (↓) | | | | | | |
|---------|------|------|------|------|------|------|------|
| | 1 | 2 | 3 | 4 | 5 | 6 | 7 |
| LiGAN | 810.45 | 981.53 | 1145.96 | 1783.95 | 1960.24 | 2547.32 | 2735.75 |
| AR | 176.67 | 222.74 | 244.51 | 268.01 | 332.89 | 388.70 | 441.90 |
| Pocket2Mol | 105.64 | 125.19 | 168.84 | 199.33 | 204.82 | 226.73 | 263.96 |
| TargetDiff | 225.48 | 253.72 | 303.60 | 344.12 | 360.74 | 420.47 | 434.30 |
| DecompDiff | 279.44 | 264.16 | 268.23 | 265.57 | 262.69 | 279.73 | 289.07 |
| DECOMPOPT | 63.33 | 169.17 | 215.19 | 248.35 | 202.81 | 237.38 | 238.32 |

## C  ADDITIONAL RESULTS

### C.1  FULL EVALUATION RESULTS

We provide box plots of evaluation metrics as shown in Figure 11.

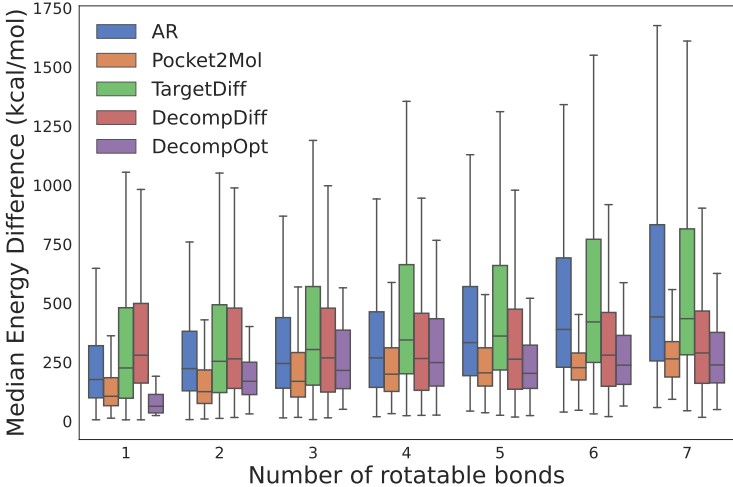

Figure 10: Median energy difference for molecules with different number of rotatable bonds before and after the force-field optimization.

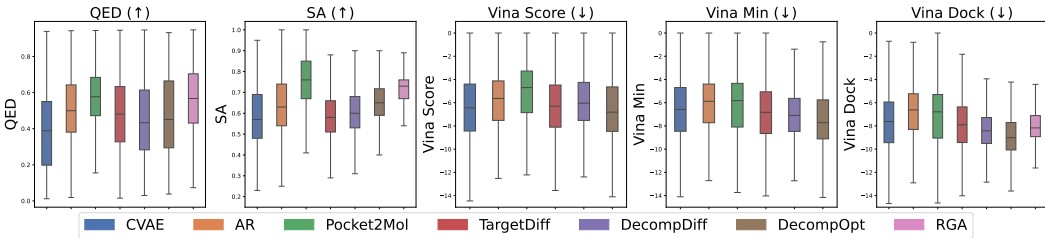

Figure 11: The boxplots of QED, SA, Vina Score, Vina Minimize, and Vina Dock of ligands generated by DECOMPOPT and baseline models.

Following Guan et al. (2023b), our model also has variants of priors. Table 9 shows the results of multiple variants of our models. The setting of *Ref Prior*, *Pocket Prior*, and *Opt Prior* strictly follows DecompDiff (Guan et al., 2023b). *Ref Best* means using the best checkpoint instead of the last checkpoint for each target pocket during optimization with reference priors for evaluation. For *Pocket/Opt Best*, it is similar. *Best of Best* means using the best checkpoint across all checkpoints with *Ref Prior* and *Pocket Prior* during optimization for each target pocket.

Table 9: Summary of different properties of reference molecules and molecules generated by our model and other generation (Gen.) and optimization (Opt.) baselines. (↑) / (↓) denotes a larger / smaller number is better.

| Methods | Vina Score (↓) | | Vina Min (↓) | | Vina Dock (↓) | | High Affinity (↑) | | QED (↑) | | SA (↑) | | Diversity (↑) | | Success Rate |
|---|---|---|---|---|---|---|---|---|---|---|---|---|---|---|---|
| | Avg. | Med. | Avg. | Med. | Avg. | Med. | Avg. | Med. | Avg. | Med. | Avg. | Med. | Avg. | Med. | |
| Reference | -6.36 | -6.46 | -6.71 | -6.49 | -7.45 | -7.26 | - | - | 0.48 | 0.47 | 0.73 | 0.74 | - | - | 25.0% |
| DECOMPOPT (Ref Prior) | -5.68 | -5.88 | -6.53 | -6.49 | -7.49 | -7.66 | 59.2% | 65.0% | 0.56 | 0.58 | 0.73 | 0.73 | 0.64 | 0.66 | 35.4% |
| DECOMPOPT (Ref Best) | -5.75 | -5.97 | -6.58 | -6.70 | -7.63 | -8.02 | 62.6% | 74.3% | 0.56 | 0.59 | 0.73 | 0.72 | 0.63 | 0.67 | 39.4% |
| DECOMPOPT (Pocket Prior) | -5.27 | -6.38 | -7.07 | -7.45 | -8.85 | -8.72 | 71.4% | 93.8% | 0.40 | 0.36 | 0.63 | 0.63 | 0.60 | 0.61 | 29.2% |
| DECOMPOPT (Pocket Best) | -5.33 | -6.49 | -7.08 | -7.60 | -9.01 | -8.98 | 73.9% | 100% | 0.41 | 0.39 | 0.63 | 0.63 | 0.59 | 0.60 | 44.7% |
| DECOMPOPT (Opt Prior) | -5.73 | -6.64 | -7.29 | -7.53 | -8.78 | -8.72 | 70.3% | 89.9% | 0.46 | 0.44 | 0.65 | 0.65 | 0.61 | 0.61 | 38.1% |
| DECOMPOPT (Opt Best) | -5.87 | -6.81 | -7.35 | -7.72 | -8.98 | -9.01 | 73.5% | 93.3% | 0.48 | 0.45 | 0.65 | 0.65 | 0.60 | 0.61 | 52.5% |
| DECOMPOPT (Best of Best) | -6.22 | -6.94 | -7.50 | -7.74 | -8.98 | -8.95 | 76.2% | 100% | 0.51 | 0.51 | 0.67 | 0.67 | 0.61 | 0.63 | 60.6% |

## C.2 TRADE-OFF BETWEEN SUCCESS RATE AND DIVERSITY

In addition to overall performance, we also show the trade-off between Success Rate and diversity of RGA, TargetDiff w/ Opt., and DECOMPOPT for each target protein pocket in Figure 12. DECOMPOPT

shows general superiority to the other two baselines in most cases considering both Success Rate and diversity.

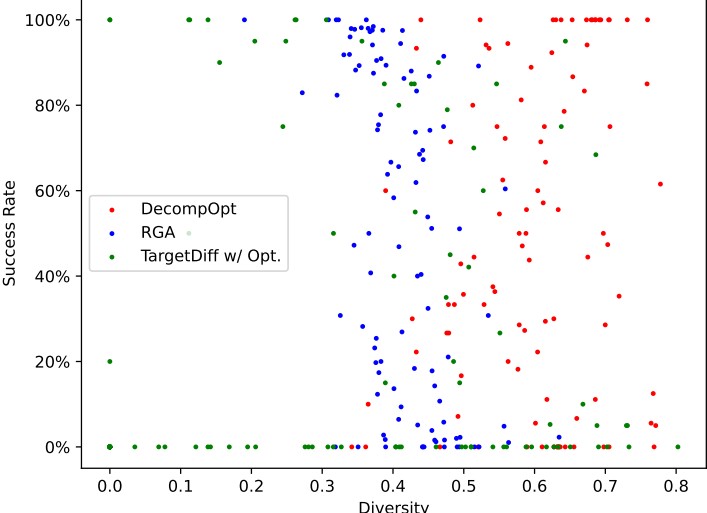

Figure 12: Trade-off of Success Rate and diversity. Each point with coordinate $(x, y)$ represents a pocket with Success Rate $x$ and diversity $y$. The closer to the top right, the better.

### C.3 EVALUATION OF THE ABILITY TO DESIGN NOVEL LIGANDS

We additionally test the **Novelty** and **Similarity** of generated ligands compared with the reference ligand. Novelty is defined as the ratio of generated ligands that are different from the reference ligands of the corresponding pockets in the test set. Similarity is defined as the Tanimoto Similarity between the generated ligands and the corresponding reference ligands. The results show that the generated ligands are not similar to reference ligands in the test set. Besides, we also test **Uniqueness** and **Diversity** of generated ligands. Uniqueness is the percentage of unique molecules among all the generated molecules. Diversity is the same as that in Section 4.1. The results are reported in Table 10. These results show that DECOMPOPT can design novel ligands, which is an important ability for drug discovery.

Table 10: Evaluation of the ability to design novel ligands.

| Methods | Novelty | Similarity | Uniqueness | Diversity |
|---|---|---|---|---|
| LiGAN | 100% | 0.22 | 87.82% | 0.66 |
| AR | 100% | 0.24 | 100% | 0.70 |
| Pocket2Mol | 100% | 0.26 | 100% | 0.69 |
| TargetDiff | 100% | 0.30 | 99.63% | 0.72 |
| DecompDiff | 100% | 0.34 | 99.99% | 0.68 |
| RGA | 100% | 0.37 | 96.82% | 0.41 |
| DECOMPOPT | 100% | 0.36 | 100% | 0.60 |

### C.4 EFFECTS OF SUBPOCKETS.

To study the influence of subpockets in controlling the optimization, we further conducted an ablation study using only arms without subpockets as conditions. As Table 11 shows, while DECOMPOPT, when solely with arms as conditions, is capable of optimizing all metrics, its efficiency in this scenario is not as well as DECOMPOPT that utilizes both arms and pockets as conditions. Recall that we use SE(3)-invaraint features of arms (and subpockets) as conditions. Without subpockets, this feature would be agnostic to the molecular interaction and spatial relation between the arms and subpockets. Such information is important to some of the properties (e.g., Vina scores). The SE(3)-invaraint

features from pairs of subpockets and arms contain the aforementioned information and are better aligned with the protein-ligand complex being generated.

Table 11: Comparison of DECOMPOPT optimization results with only arms and arm-pocket complexes as conditions. (↑) / (↓) denotes a larger / smaller number is better.

| Methods | Vina Score (↓) | | Vina Min (↓) | | Vina Dock (↓) | | High Affinity (↑) | | QED (↑) | | SA (↑) | | Diversity (↑) | | Success Rate |
|---|---|---|---|---|---|---|---|---|---|---|---|---|---|---|---|
| | Avg. | Med. | Avg. | Med. | Avg. | Med. | Avg. | Med. | Avg. | Med. | Avg. | Med. | Avg. | Med. | |
| DecompDiff | -5.67 | -6.04 | -7.04 | -7.09 | -8.39 | -8.43 | 64.4% | 71.0% | 0.45 | 0.43 | 0.61 | 0.60 | 0.68 | 0.68 | 24.5% |
| DECOMPOPT (arms-only) | -5.52 | -6.26 | -7.05 | -7.26 | -8.65 | -8.64 | 66.6% | 86.1% | 0.46 | 0.43 | 0.63 | 0.63 | 0.63 | 0.63 | 45.7% |
| DECOMPOPT | -5.87 | -6.81 | -7.35 | -7.72 | -8.98 | -9.01 | 73.5% | 93.3% | 0.48 | 0.45 | 0.65 | 0.65 | 0.60 | 0.61 | 52.5% |

## C.5 INFLUENCE OF THE QUALITY OF INITIAL LIGANDS ON PERFORMANCE.

To study the influence of the quality of initial ligands on performance of structure-based molecular optimization, we have conducted an ablation study focusing on Vina Min score optimization, using ligands with high and low Vina Min scores as initializations for the arm lists. Due to limited resources, we chose to conduct this study on the protein 2V3R, which is randomly chosen from our test set. We generated 100 ligands using DecompDiff and selected 20 ligands with the highest and lowest Vina Min scores. These ligands were then used as the initial conditions for the optimization process. As shown in Table 12, the optimization outcomes are slightly influenced by the quality of the initial ligands. However, regardless the quality of the initial ligands, DECOMPOPT can consistently improve the quality of the generated ligands.

Table 12: Comparison of optimization with initial ligands of different quality.

| | High Vina Min Scores | | Low Vina Min Scores | | Δ (high - low) |
|---|---|---|---|---|---|
| | Avg. | Med. | Avg. | Med. | Avg. |
| Initial ligands | -8.54 | -8.48 | -7.08 | -7.04 | -1.46 |
| DECOMPOPT | -9.12 | -8.96 | -9.00 | -8.96 | -0.12 |

## C.6 INFLUENCE OF THE NUMBER OF INITIAL LIGANDS ON PERFORMANCE.

To study the influence of the number of initial molecules on the performance of structure-based molecular optimization, we further run the experiments with initial arm lists of 1 and 5 molecules generated by DecompDiff. As Table 13 indicates, the initial number of molecules has a modest impact on the optimization outcomes, with a higher number of molecules generally leading to improved performance. Notably, even when starting with a single molecule generated by DecompDiff, DECOMPOPT demonstrates a considerably high success rate.

Table 13: Summary of results using different number of molecules to initialize arm lists. (↑) / (↓) denotes a larger / smaller number is better.

| Methods | Vina Score (↓) | | Vina Min (↓) | | Vina Dock (↓) | | High Affinity (↑) | | QED (↑) | | SA (↑) | | Diversity (↑) | | Success Rate |
|---|---|---|---|---|---|---|---|---|---|---|---|---|---|---|---|
| | Avg. | Med. | Avg. | Med. | Avg. | Med. | Avg. | Med. | Avg. | Med. | Avg. | Med. | Avg. | Med. | |
| init num = 1 | -5.41 | -6.61 | -7.12 | -7.51 | -8.78 | -8.82 | 70.4% | 88.9% | 0.47 | 0.45 | 0.64 | 0.63 | 0.61 | 0.61 | 47.0% |
| init num = 5 | -5.71 | -6.71 | -7.25 | -7.58 | -8.86 | -8.97 | 71.8% | 93.3% | 0.49 | 0.46 | 0.65 | 0.64 | 0.60 | 0.61 | 49.4% |
| init num = 20 | -5.87 | -6.81 | -7.35 | -7.72 | -8.98 | -9.01 | 73.5% | 93.3% | 0.48 | 0.45 | 0.65 | 0.65 | 0.60 | 0.61 | 52.5% |

# D EXTENDED RESULTS OF CONTROLLABILITY

## D.1 R-GROUP OPTIMIZATION

We provide additional R-group Optimization experiment on protein 4G3D, as shown in Figure 13.

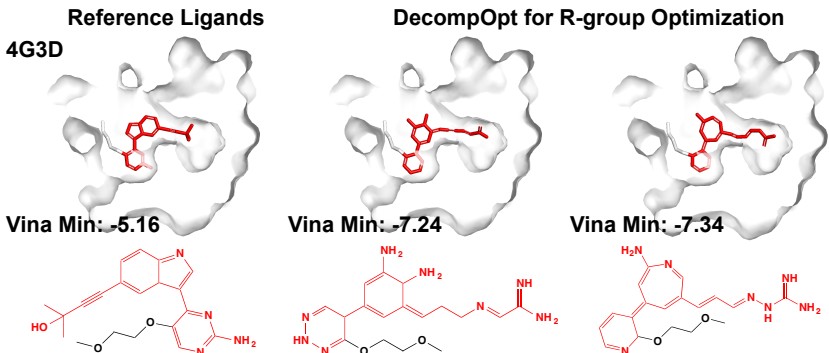

Figure 13: Additional R-group optimization result. The left column is reference binding molecule, the middle and right columns are molecules generated by DECOMPOPT with 30 rounds of optimization on protein 4G3D. Optimized R-group are highlighted in red.

Table 14: R-group optimization results generated using Decompdiff and DECOMPOPT on protein 3DAF and 4F1M. DECOMPOPT was optimized over 30 rounds towards high Vina Min Score and evaluated using the final round results. Both targets were assessed with 20 generated molecules and the mean of properties are reported.

| Model | 3DAF | | | 4F1M | | |
|---|---|---|---|---|---|---|
| | Vian Min ($\downarrow$) | Tanimoto Sim. ($\uparrow$) | Complete. ($\uparrow$) | Vian Min ($\downarrow$) | Tanimoto Sim. ($\uparrow$) | Complete. ($\uparrow$) |
| Decompdiff | -8.44 | 0.15 | 60.0% | -5.90 | 0.15 | 65.0% |
| DECOMPOPT | -9.39 | 0.23 | 95.0% | -6.32 | 0.49 | 55.0% |

## D.2 FRAGMENT GROWING

Enhancing the binding affinity of drug candidates through combination of R-group optimization and fragment growing can effectively leverage capabilities of DECOMPOPT. The quantitative results are shown in Table 14. For our case study, we perform R-group optimization and fragment growing on 5AEH. Starting from a high binding affinity drug candidate, we first optimize R-group for 30 rounds same as workflow in Section 4. Subsequently, we design the new arms prior and atom num with expert guidance, and expand fragments using DECOMPOPT. As Figure 14 shows, DECOMPOPT ultimately generates molecules with a Vina Min Score more than 4 kcal/mol better than the reference.

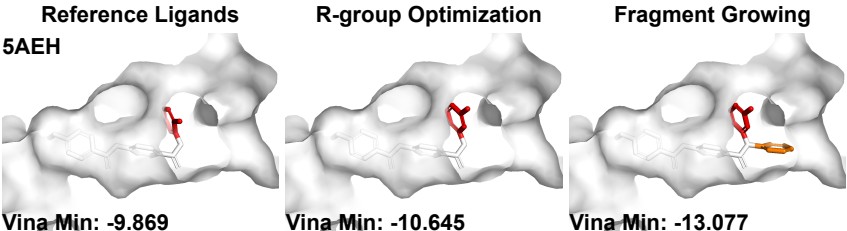

Figure 14: Example of R-group optimization and fragment growing conducted using DECOMPOPT on 5AEH. The reference ligand, the best R-group result, and the best fragment growing result based on R-group optimization are displayed from left to right. The selected R-group is highlighted in red, while the newly extended arm is highlighted in orange.

## D.3 SCAFFOLD HOPPING

**Additional Evaluation Metrics** In addition to evaluation metrics discussed in Section 4, we evaluated *Validity*, *Uniqueness*, *Novelty*, *Complete Rate*, and *Scaffold Similarity* to measure models' capability in scaffold hopping. Detailed calculation of these metrics as follows:

- **Validity** is defined as the fraction of generated molecules that can be successfully sanitized.

- **Uniqueness** measures the proportion of unique molecules among the generated molecules.

- **Novelty** measures the fraction of generated molecules that not presented in training set.

- **Complete Rate** measures the proportion of completed molecules within the generated results.

- **Scaffold Similarity** Following Polykovskiy et al. (2020), Bemis–Murcko scaffolds are extracted using rdkit function `MurckoScaffold`. We count the occurrences of scaffolds in all generated and reference molecules, creating vectors $G$ and $R$, where each dimension represents the count of a specific scaffold. The scaffold similarity is calculated as the cosine similarity between vectors $G$ and $R$.

**More Examples of Generated Results** For scaffold hopping, we provide more visualization of ligands generated by DECOMPOPT and DecompDiff on protein 2Z3H, 4AVW, 4QLK, and 4BEL, which are shown in Figure 15.

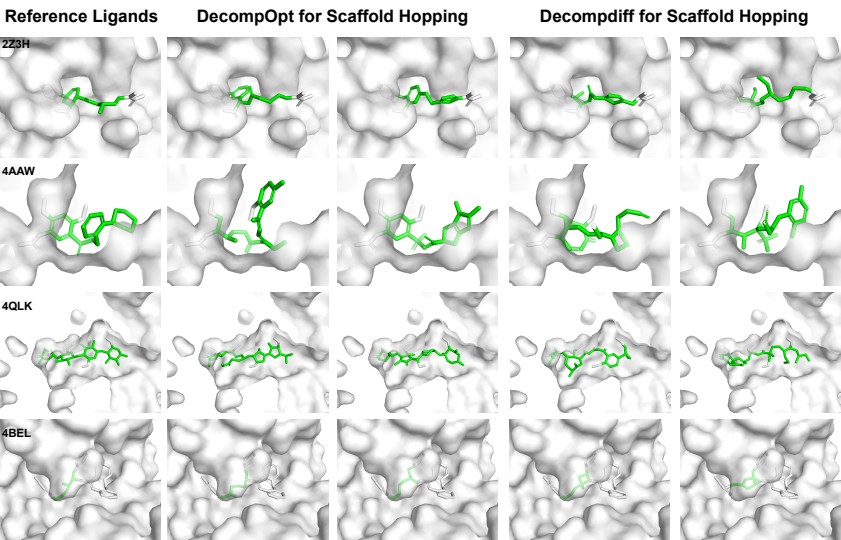

Figure 15: More examples of Scaffold Hopping results. The left column shows reference ligands, Scaffold Hopping results generated by DECOMPOPT are shown at the second and the third rows, and results generated by DecompDiff are shown at the fourth and the fifth rows. Scaffold are highlighted in green.

**Changes in Molecular Properties After Scaffold Hopping** Scaffold hopping aims at finding scaffold structures that can connect existing functional groups without disrupting their interactions with the target protein. The main purpose of this is to find novel scaffolds which are not protected by existing patents while maintaining comparable properties as the original molecule. Therefore, we did not implement property optimization mechanisms in scaffold hopping tasks and solely focusing on designing scaffolds that can connect existing arms. We provide the property comparison before and after scaffold hopping in Table 15. As the result shows, the properties of the ligands remain relatively consistent before and after the process of scaffold hopping.

Table 15: Summary of properties of reference molecules and molecules generated through scaffold hopping using DECOMPOPT. ($\uparrow$) / ($\downarrow$) denotes a larger / smaller number is better.

| Methods | Vina Score ($\downarrow$) | | Vina Min ($\downarrow$) | | Vina Dock ($\downarrow$) | | QED ($\uparrow$) | | SA ($\uparrow$) | |
|---|---|---|---|---|---|---|---|---|---|---|
| | Avg. | Med. | Avg. | Med. | Avg. | Med. | Avg. | Med. | Avg. | Med. |
| Reference | -6.36 | -6.46 | -6.71 | -6.49 | -7.45 | -7.26 | 0.48 | 0.47 | 0.73 | 0.74 |
| Scaffold Hopping by DECOMPOPT | -5.89 | -6.13 | -6.46 | -6.28 | -7.28 | -7.48 | 0.49 | 0.48 | 0.71 | 0.69 |

