# OpenReview forum: "DecompOpt: Controllable and Decomposed Diffusion Models for Structure-based Molecular Optimization"
_ICLR.cc/2024/Conference — ICLR 2024 poster_

### Official Review · Reviewer_dt3J · 2023-10-25

**Soundness:** 2 fair
**Presentation:** 2 fair
**Contribution:** 3 good
**Rating:** 6
**Confidence:** 4

**Summary:**

The paper proposes DecompOpt, a framework for de novo design and optimisation of ligands in protein binding sites conditioned on the pocket and a set of templates for arms and scaffold. The method is trained on the CrossDocked2020 benchmark and applied to de novo design, R-group optimisation and scaffold hopping.

**Strengths:**

- DecompOpt is a flexible framework that is applicable to practical use cases in drug design
- SOTA results on CrossDocked2020 benchmark

**Weaknesses:**

- Confidence intervals in evaluation are missing.
- The overall clarity of the text should be improved (typos/grammar/…).
- The complex framework is not described very well and is hard to grasp. For example, the “model architecture, training loss etc” is taken from DecompDiff and only referenced. An overview figure might help to better show the overall framework and identify novel contributions.
- It is not clear what the arm / subpocket conditioning exactly encodes: Is this just guiding towards similar arms, or is the sub-pocket important here? An additional ablation using only the arms, A_k = Enc(A_k), would be informative.
- It is unclear / not evaluated, how much the method relies on the availability of high-quality reference ligands vs the generative capabilities of the diffusion model. For example, a simple 2d baseline that recombines arms of different reference structures might already perform very well if already a large set of references is available.

**Questions:**

- Criteria for success rate seem arbitrary. How were these thresholds chosen?
- Where does the difference for the DecompDiff results between Tabs 1 and 3 stem from?
- Is DecompOpt applicable if no/only few reference ligands are available? How does the performance depend on the number / performance of initial reference ligands?

Update:
I thank the authors for the extensive answers to my questions and the additional experiments and clarifications. I raised my score accordingly.

---

> ### Author Response · Authors · 2023-11-21
> **Response to Reviewr dt3J (1/2)**
>
> Thank you for your constructive comments. Please see below for our responses to the comments.
>
> **Q1: "Confidence intervals in evaluation are missing."**
>
> A1: We have the confidence intervals in the appendix in the revision. Please refer to Appendix C.1 for the boxplot of the evaluation metrics.
>
> We have conducted paired sample t-test for success rate of DecompOpt and its counterpart, DecompDiff. The p-value of the success rate difference between DecompOpt and DecompDiff is $1.83\times 10^{-17}$, which shows that the optimization works as we expected and demonstrates the significant superiority of our method.
>
> The results show that the superiority of our method is consistent.
>
> **Q2: "The overall clarity of the text should be improved (typos/grammar/…)."**
>
> A2: Thanks for pointing this out! We will carefully proofread our manuscript and fix all the typos.
>
> **Q3: The complex framework is not described very well and is hard to grasp, such as the model architecture and loss function. "An overview figure might help to better show the overall framework and identify novel contributions."**
>
> A3: We have added more explanation of the model design and a more detailed overview figure in Appendix A.
>
> **Q4: "Is this just guiding towards similar arms, or is the sub-pocket important here?  An additional ablation using only the arms, A_k = Enc(A_k), would be informative."**
>
> A4: Thanks for your insightful questions. The results with only arms and without subpockets are as follows. Recall that we use SE(3)-invariant features of arms (and subpockets) as conditions. Without subpockets, this feature would be agnostic to the molecular interaction and spatial relation between the arms and subpockets. Such information is important to some of the properties (e.g., Vina scores). The SE(3)-invariant features from pairs of subpockets and arms contain the aforementioned information and are better aligned with the protein-ligand complex being generated. The results indicate that DecompOpt, when solely with arms as conditions, is capable of optimizing all metrics. However, its efficiency in this scenario is not as well as DecompOpt that utilizes both arms and pockets as conditions.
>
> |  | Vina Score (Avg.) | Vina Score (Med.) | Vina Min (Avg.) | Vina Min (Med.) | Vina Dock (Avg.) | Vina Dock (Med.) | High Affinity (Avg.) | High Affinity (Med.) | QED (Avg.) | QED (Med.) | SA (Avg.) | SA (Med.) | Diversity (Avg.) | Diversity (Med.) | Success Rate (Avg.) |
> | --- | --- | --- | --- | --- | --- | --- | --- | --- | --- | --- | --- | --- | --- | --- | --- |
> | DecompDiff | -5.67 | -6.04 | -7.04 | -7.09 | -8.39 | -8.43 | 64.4% | 71.0% | 0.45 | 0.43 | 0.61 | 0.60 | 0.68 |  0.68 | 24.5% |
> | DecompOpt (arms-only) | -5.52 | -6.26 | -7.05 | -7.26 | -8.65 | -8.64 | 66.6% | 86.1% | 0.46 | 0.43 | 0.63 | 0.63 | 0.63 | 0.63 | 45.7% |
> | DecompOpt | -5.87  | -6.81  | -7.35  | -7.72  | -8.98  | -9.01 | 73.5%   | 93.3% | 0.48   | 0.45 | 0.65  | 0.65 | 0.60  | 0.61 | 52.5% |
>
> **Q5: It is unclear / not evaluated how much the method relies on the availability of high-quality reference ligands vs the generative capabilities of the diffusion model.**
>
> A5: We believe there might be a misunderstanding here. We do not use the reference ligands in the test set as conditions. Instead, we initialize the conditional arm lists with ligands generated by DecompDiff. We also ablate on how the quality of the initial ligands impacts the performance of DecompOpt.
>
> We conducted an ablation study focusing on Vina Min score optimization, using ligands with high and low Vina Min scores as initializations for the arm lists. Due to limited resources, we chose to conduct this study on the protein 2v3r, which is randomly chosen from our test set. We generated 100 ligands using DecompDiff and selected 20 ligands with the highest and lowest Vina Min scores. These ligands were then used as the initial conditions for the optimization process.
>
> |  | High Vina Min (Avg.) | High Vina Min (Med.) | Low Vina Min (Avg.) | Low Vina Min (Med.) | $\Delta$ (high - low) (Avg.) |
> | --- | --- | --- | --- | --- | --- |
> | initial ligands | -8.54 | -8.48 | -7.08 | -7.04 | -1.46 |
> | DecompOpt | -9.12 | -8.96 | -9.00 | -8.96 | -0.12 |
>
> As shown in the experiment result, the optimization outcomes are slightly influenced by the quality of the initial ligands. However, regardless of the quality of the initial ligands, DecompOpt can consistently improve the quality of the generated ligands.

---

> ### Author Response · Authors · 2023-11-21
> **Response to Reviewr dt3J (2/2)**
>
> **Q6: "Criteria for success rate seem arbitrary. How were these thresholds chosen?"**
>
> A6: We want to clarify that the thresholds of QED, SA, and Vina as criteria for success rate are not arbitrary but instead traceable. The Vina threshold (< -8.18 kcal/mol) corresponds to 1uM binding affinity, and it is a common requirement for a potential drug candidate in practical drug discovery; the QED and SA thresholds are calculated as the 10th percentile of DrugCentral [1] (a database of up-to-date drugs and pharmaceuticals), to reflect the latest drug property distribution. These thresholds are also used by [2] and [3].
>
> **Q7: "Where does the difference for the DecompDiff results between Tabs 1 and 3 stem from?"**
>
> A7: Following DecompDiff, we use different decomposed priors for ablation studies. Opt Prior is used in Tab 1 and Ref Prior is used in Tab 3. Different kinds of decomposed priors are defined by DecompDiff.
>
> _Ref Prior_ is calculated using a Gaussian distribution based on the reference molecule, determined through maximum likelihood estimation. _Pocket Prior_ employs AlphaSpace2 [4] for identifying subpockets, which is used for estimating the prior center. Additionally, it uses a neural classifier to estimate the number of ligand atoms and the prior's standard deviation. _Opt Prior_ adopts Ref Prior when the reference ligand meets the required success criteria. Otherwise, it employs the Pocket Prior.
>
> **Q8: "Is DecompOpt applicable if no/only few reference ligands are available? How does the performance depend on the number / performance of initial reference ligands?"**
>
> Thank you for your question. As clarified in Q5, reference ligands are not used as initial conditions for DecompOpt. Sample results provided in Table 1 initialize the arm lists from 20 ligands generated by DecompDiff. We further run the experiments with initial the arm lists of 1 and 5 ligands generated by DecompDiff. The results are shown as follows.
>
> | Init num | Vina Score (Avg.) | Vina Score (Med.) | Vina Min (Avg.) | Vina Min (Med.) | Vina Dock (Avg.) | Vina Dock (Med.) | High Affinity (Avg.) | High Affinity (Med.) | QED (Avg.) | QED (Med.) | SA (Avg.) | SA (Med.) | Diversity (Avg.) | Diversity (Med.) | Success Rate (Avg.) |
> | --- | --- | --- | --- | --- | --- | --- | --- | --- | --- | --- | --- | --- | --- | --- | --- |
> | 1 | -5.41 | -6.61 | -7.12 | -7.51 | -8.78 | -8.82 | 70.4% | 88.9% | 0.47 | 0.45 | 0.64 | 0.63 | 0.61 | 0.61 | 47.0% |
> |  5 | -5.71 | -6.71 | -7.25 | -7.58 | -8.86 | -8.97 | 71.8% | 93.3% | 0.49 | 0.46 | 0.65 | 0.64 | 0.60 | 0.61 | 49.4% |
> |  20 | -5.87  | -6.81  | -7.35  | -7.72  | -8.98  | -9.01 | 73.5%  | 93.3% | 0.48  | 0.45 | 0.65  | 0.65 | 0.60  | 0.61 | 52.5% |
>
> The results indicate that the initial number of ligands has a modest impact on the optimization outcomes, with a higher number of ligands generally leading to improved performance. Notably, even when starting with a single ligand generated by DecompDiff, DecompOpt demonstrates a higher success rate compared to other baseline models.
>
> **References:**
>
> [1] Ursu, Oleg, et al. DrugCentral: online drug compendium. Nucleic acids research (2016).
>
> [2] Long, Siyu, et al. Zero-Shot 3D Drug Design by Sketching and Generating. NeurIPS (2022).
>
> [3] Guan, Jiaqi, et al. DecompDiff: Diffusion Models with Decomposed Priors for Structure-Based Drug Design. ICML (2023).
>
> [4] Katigbak, Joseph, et al. AlphaSpace 2.0: representing concave biomolecular surfaces using β-clusters. Journal of chemical information and modeling (2020).

---

> ### Author Response · Authors · 2023-11-22
> **Thanks for your positive feedback!**
>
> Thank you again for your positive feedback and recognition of our efforts!

---

### Official Review · Reviewer_ewuH · 2023-10-29

**Soundness:** 2 fair
**Presentation:** 3 good
**Contribution:** 2 fair
**Rating:** 5
**Confidence:** 4

**Summary:**

This paper aims to address the problem of generating 3D molecules with desired properties by introducing a controllable and decomposed diffusion model. Specifically, this paper proposes a model DECOMPOPT, which learns molecular grammar by iterative molecular optimization.The approach decomposes the molecules into scaffolds and arms, aiming to replace underperforming arms with new arms that exhibit improved properties.

**Strengths:**

1.	The paper is well-written and easy to follow.
2.	The experiment results are comprehensive.

**Weaknesses:**

1. My main concern regarding this article is the lack of alignment between the motivation and the approach. While the article acknowledges the importance of the property, it doesn't directly address it.

   a) Which part of the article is designed to address QED or easily synthesizable compounds?

   b) The Vina score from Table 1 appears to be subpar. Although I understand that the Vina score may not be the most suitable metric due to its weak correlation with affinity, the authors use it for optimization and as an indicator for data analysis(Figure 1). This raises the question of why they emphasize the results of Vina min and but not reporting the median or some other indicator.

2. I am curious about the inference efficiency of the diffusion model. Since diffusion models are not known for their efficiency during inference, could the authors compare the generation efficiency of the proposed method with that of other models?

3. Did the authors evaluate the rationality of the generated conformations? For instance, did they assess the energy difference before and after force field optimization? As diffusion-based models like DecompDiff has problems with this.

4. The second case in Figure 3 seems questionable or may be difficult to interpret. I recommend redrawing it for clarity.

5. Scaffold hopping is an interesting aspect, but the authors have not considered the potential changes in molecular properties (vina score, for example) after the hopping process.

**Questions:**

(Updated) Thank you for providing additional experimental details and clarification. I appreciate that my main concerns have been mostly addressed, especially the results for conformation optimization. I would raise the score from 3 to 5.

---

> ### Author Response · Authors · 2023-11-20
> **Response to Reviewer ewuH (1/2)**
>
> Thank you for your feedback. Please see below for our responses to the comments.
>
> **Q1: Misalignment between the motivation and the approach. The importance of property is not directly addressed. "Which part of the article is designed to address QED or easily synthesizable compounds?"**
>
> A1: We would like to clarify that our model, DecompOpt, while property-agnostic in its core design, is adeptly complemented by a property-oriented optimization process. This combination is key to the success of our method and facilitates the generation of compounds with specifically desired properties such as QED and SA.
>
> The iterative optimization is achieved by high-quality arm lists, which are curated using a z-score metric. As described in Section 4.1 (Implementation Details), this metric comprehensively evaluates QED, SA, and Vina Min, ensuring that the optimization process is aligned with the desired molecular properties. So our motivation and our approach are actually quite aligned.
>
> **Q2: "The Vina score from Table 1 appears to be subpar." The motivation for using Vina scores for optimization and data analysis (Figure 1). Why are the results of Vina min emphasized but not the median or some other indicators?**
>
> A2: Thank you for bringing up this concern. Indeed, this brings up a crucial aspect that merits clarification. To properly articulate our approach, it is essential to first distinguish between Vina Score, Vina Min, and Vina Dock, all computed by AutoDock Vina software.
> - Vina Score predicts the binding affinity of a ligand to a target protein directly.
> - Vina Min is calculated by a process where, after an initial docking prediction, further subtle refinement is done using force field-based optimization around the binding site.
> - Vina Dock is obtained after re-docking a ligand into a protein's binding site.
>
> All the three metrics are commonly used in estimating binding affinity in recent SBDD studies. We chose Vina Min as our optimization objective due to its capacity to reflect potential binding affinity with local optimization, aligning with our method’s focus on arm optimization without incurring a high time cost. Other binding affinity indicators can also be easily adapted to our property-oriented optimization process.
>
> Regarding the CrossDocked2020 dataset, it is constructed from the PDBBind database through cross-docking ligands and proteins. The Vina Score shown in Figure 1 reflects the binding affinity of docked poses in CrossDocked2020.
>
> Regarding the subpar Vina Score for DecompOpt in Table 1, it is important to note that while combining TargetDiff with our optimization process may yield superior Vina scores, this comes at the cost of reduced molecular diversity. DecompOpt effectively balances optimization objectives with diversity, leveraging the flexibility provided by the scaffold structure and diffusion process. In contrast, TargetDiff w/ Opt tends to generate ligands similar to the initial ligands, which is not ideal in drug design due to the need for diverse molecular candidates. To comprehensively evaluate the quality of generated ligands, metrics such as success rate, which was also used in [1,2], are more important.
>
> **Q3: Inference efficiency of the diffusion model. Generation efficiency compared with other methods.**
>
> A3: For inference efficiency, we test the run time for generating 20 molecules on average. In detail, AR, Pocket2Mol, TargetDiff, DecompDiff, and DecompOpt use 826s, 269s, 367s, 506s, and 731s, respectively. Our model is on par with some of the baselines. The inference is efficient and totally acceptable in practice of drug discovery.

---

> ### Author Response · Authors · 2023-11-20
> **Response to Reviewer ewuH (2/2)**
>
> **Q4: Evaluation of rationality of the generated conformations. The energy difference before and after force field optimization.**
>
> A4:
> Thanks for your valuable feedback!
>
> We have already evaluated the rationality of generated conformations in Appendix B. More specifically, we compute atom-pair distance, different bond distance, different angle distributions of generated molecules and compare them against the corresponding reference empirical distributions. In most cases, our model performs better than all of the baselines.
>
> Following your suggestions, we also evaluate energy difference (kcal/mol). We optimize the generated structures with Merck Molecular Force Field (MMFF) [3] and calculate the energy difference between pre- and pos- MMFF-optimized coordinates for different rigid fragments (consisting of 3/4/5/6/7/8 atoms) that do not contain any rotatable bonds. The results are as follows:
>
> | Methods | 3 | 4 | 5 | 6 | 7 | 8 |
> | --- | --- | --- | --- | --- | --- | --- |
> | liGAN | 86.32 | 165.15 | 105.96 | 185.70 | 243.79 | 332.81 |
> | AR | 25.79 | 73.06 | 23.89 | 30.42 | 56.47 | 76.50 |
> | Pocket2Mol | 10.43 | 33.93 | 34.47 | 27.86 | 33.90 | 42.97 |
> | TargetDiff | 7.31 | 30.57 | 18.01 | 11.98 | 28.92 | 50.42 |
> | DecompDiff | 6.01 | 29.20 | 10.78 | 4.33 | 12.74 | 30.68 |
> | DecompOpt | 6.00 | 16.59 | 9.89 | 2.61 | 13.29 | 31.49 |
>
> As the results indicate, DecompOpt achieves low energy differences and outperforms baselines in most cases.
>
> We also calculate the energy difference before and after force field optimization for the whole molecules with 1/2/3/4/5/6/7 rotatable bonds respectively. The results are as follows:
>
> | Methods | 1 | 2 | 3 | 4 | 5 | 6 | 7 |
> | --- | --- | --- | --- | --- | --- | --- | --- |
> | liGAN | 810.45 | 981.53 | 1145.96 | 1783.95 | 1960.24 | 2547.32 | 2735.75 |
> | AR | 176.67 | 222.74 | 244.51 | 268.01 | 332.89 | 388.70 | 441.90 |
> | Pocket2Mol | 105.64 | 125.19 | 168.84 | 199.33 | 204.82 | 226.73 | 263.96 |
> | TargetDiff | 225.48 | 253.72 | 303.60 | 344.12 | 360.74 | 420.47 | 434.30 |
> | DecompDiff | 279.44 | 264.16 | 268.23 | 265.57 | 262.69 | 279.73 | 289.07 |
> | DecompOpt | 63.33 | 169.17 | 215.19 | 248.35 | 202.81 | 237.38 | 238.32 |
>
> Notably, although we do not explicitly introduce optimization over conformation, DecompOpt outperforms the other diffusion-based models by a large margin.
>
> We also show the corresponding box plot in Appendix B in the revision. These results show that the conformation of ligands generated by DECOMPOPT is high-quality and stable.
>
> **Q5: “The second case in Figure 3 seems questionable or may be difficult to interpret.”**
>
> A5: Thank you for your suggestion. In R-group optimization, we iteratively modify R-groups for higher Vina Min scores, leading to improved binding affinities. While the changes in atoms and bond types might appear subtle, their effects with positional adjustments significantly impact binding affinity, as demonstrated in Figure 3. For example, for ligands of protein 4F1M (see the bottom left and bottom right of Figure 3), the Carbon atom connecting two Phosphorus atoms in R-group of the reference ligand is replaced with the Oxygen atom in the optimized ligands. We redraw Figure 3 in the revision for clarity. For broader validation, we've also conducted additional experiments on protein 4G3D, which is provided in Appendix D.1.
>
> **Q6: Changes in molecular properties (vina score, for example) after the hopping process.**
>
> A6: Scaffold hopping aims at finding scaffold structures that can connect existing functional groups without disrupting their interactions with the target protein. The main purpose of this is to find novel scaffolds which are not protected by existing patents while maintaining comparable properties as the original molecule. Therefore, we did not implement property optimization mechanisms in scaffold hopping tasks and solely focused on designing scaffolds that can connect existing arms. We provide the property comparison before and after scaffold hopping in the table below.
>
> |  | Vina Score (Avg.) | Vina Score (Med.) | Vina Min (Avg.) | Vina Min (Med.) | Vina Dock (Avg.) | Vina Dock (Med.) | QED (Avg.) | QED (Med.) | SA (Avg.) | SA (Med.) |
> | --- | --- | --- | --- | --- | --- | --- | --- | --- | --- | --- |
> | Reference ligands | -6.36 | -6.46 | -6.71 | -6.49 | -7.45 | -7.26 | 0.48 | 0.47 | 0.73 | 0.74 |
> | DecompOpt | -5.89 | -6.13 | -6.46 | -6.28 | -7.28 | -7.48 | 0.49 | 0.48 | 0.71 | 0.69 |
>
> The evaluation results demonstrate that the properties of the ligands remain relatively consistent before and after the process of scaffold hopping.
>
> **References:**
>
> [1] Long, Siyu, et al. Zero-Shot 3D Drug Design by Sketching and Generating. NeurIPS (2022).
>
> [2] Guan, Jiaqi, et al. DecompDiff: Diffusion Models with Decomposed Priors for Structure-Based Drug Design. ICML (2023).
>
> [3] Halgren, Thomas A. Merck molecular force field. I. Basis, form, scope, parameterization, and performance of MMFF94. Journal of computational chemistry (1996).

---

> ### Author Response · Authors · 2023-11-22
> **Gentle Reminder**
>
> Dear Reviewer ewuH,
>
> Thanks again for your valuable feedback! We sincerely appreciate the time and effort you have dedicated to reviewing our paper.
>
> To respond to your comments, we have made more detailed explanations of our motivation and method, and also done extensive experiments and comprehensive evaluation.
>
> As the discussion phase will end tomorrow, we kindly request, if possible, that you review our rebuttal at your earliest convenience. If you have any other concerns, we would like to further discuss. If we have addressed your concerns, we sincerely hope you can reconsider the evaluation of our paper.

---

> ### Author Response · Authors · 2023-11-23
> **Follow up**
>
> Dear Reviewer ewuH,
>
> We kindly remind you that the author-reviewer discussion phase will end soon.
>
> We have revised our paper accordingly and made more detailed explanations of our motivation and method, as well as extensive experiments and comprehensive evaluation.
>
> We look forward to your feedback about our response and revision so that we can further address your other concerns if there are any.
>
> And we sincerely hope you can reconsider the evaluation of our submission if we have addressed your concerns.

---

### Official Review · Reviewer_SQ7r · 2023-11-01

**Soundness:** 3 good
**Presentation:** 3 good
**Contribution:** 2 fair
**Rating:** 6
**Confidence:** 3

**Summary:**

The paper focuses on the task of conditional generation in structure-based drug design, aiming to generate drugs with specific properties. The proposed method is an optimization approach based on a controllable and decomposed diffusion model, offering fine-grained control and local optimization. The method introduces a mechanism for precise control over the arms of the generated ligands. Subsequently, the ligand molecules, including atom coordinates, atom type, and bond type, can be generated by conditioning on certain arms and the target pockets using the diffusion model. Once a ligand molecule is derived, it is treated as a reference molecule and further optimized using evolutionary algorithms. The author presents the results from two perspectives: the generation perspective and the optimization perspective  to evaluate the target binding affinity and molecular properties.

**Strengths:**

The paper gives a decomposed method on drug design, generation on arms and scaffold. DECOMPOPT combines the advantages of diffusion models and optimization algorithms, utilizing diffusion models to extract molecular grammar and optimization algorithms to effectively optimize desired properties.

**Weaknesses:**

The method section containing diffusion part also used in the previous tasks while the author does not point out their novelty. The implementation detail is hard to reproduce

**Questions:**

1. Why can solely maximizing the likelihood of training data mislead SBDD models?
2. What is your novelty in section 3.1 comparing to DecompDiff?
3. How is diversity achieved in the generated ligand molecules without explicitly introducing regularization over the representation space of conditions?
4. Have you try multiple time experiments with differnet random seeds to avoid the randomness?

---

> ### Author Response · Authors · 2023-11-20
> **Response to Reviewer SQ7r**
>
> Thank you for your constructive feedback. Please see below for our responses to the comments.
>
> **Q1: Novelty of the diffusion part compared with DecompDiff. Reproducibility.**
>
> A1: DecompDiff incorporates decomposition into diffusion models for SBDD by introducing the decomposed prior, which only contains spatial information. In contrast, DecompOpt introduces reference arms as control signals to leverage both chemical and spatial semantics from discovered molecular substructures. This makes DecompOpt fully utilize the advantages of decomposition. Moreover, thanks to controllability, we can effectively achieve molecular optimization and controllable molecular generation based on the model, while DecompDiff cannot.
>
> As for reproducibility, we are committed to releasing the codes, model checkpoints, and samples generated by our models upon acceptance. Besides, we have provided more implementation details in Appendix A in the revision.
>
> **Q2: "Why can solely maximizing the likelihood of training data mislead SBDD models?"**
>
> A2: Thanks for your insightful question. As we observed, the training data (e.g., CrossDocked2020, a widely used dataset) is not satisfying in terms of the desired properties. As shown in Figure 1, more than half of the ligands in the dataset are under the desired threshold, which means they may not bind to the targets perfectly. Therefore, the performance of the generative models that solely maximize the likelihood cannot satisfy the criteria of drug discovery. This motivates us to introduce molecular optimization into generative models.
>
> **Q3: "How is diversity achieved in the generated ligand molecules without explicitly introducing regularization over the representation space of conditions?"**
>
> A3: The diversity comes from the randomness in diffusion sampling and the dedicated design of our method. Though we provide the model with conditions on arms to achieve controllability, we leave room for exploring scaffolds. Since the atoms of the arms and scaffold are correlated during sampling,  the randomness is also introduced to the arms. That is also the reason why the generated arms are similar to rather than exactly the same as the arms provided as conditions. Even in the absence of explicitly introducing regularization such as VAE over the representation space, a significant level of diversity can still be attained.
>
> **Q4: "Have you tried multiple time experiments with different random seeds to avoid randomness?"**
>
> A4: Following prior works, all results reported in tables are mean or median values of properties of generated ligands over 100 pockets. Per your suggestion, we furthermore run the experiments under different random seeds. The results are shown as follows:
>
> | seed | Vina Score (Avg.) | Vina Score (Med.) | Vina Min (Avg.)  | Vina Min (Med.) | Vina Dock (Avg.) | Vina Dock (Med.) | High Affinity (Avg.) | High Affinity (Med.) | QED (Avg.) | QED (Med.) | SA (Avg.) | SA (Med.) | Diversity (Avg.) | Diversity (Med.) | Success Rate (Avg.) |
> | --- | --- | --- | --- | --- | --- | --- | --- | --- | --- | --- | --- | --- | --- | --- | --- |
> | - | -5.87  | -6.81  | -7.35  | -7.72  | -8.98  | -9.01 | 73.5%   | 93.3% | 0.48   | 0.45 | 0.65  | 0.65 | 0.60  | 0.61 | 52.5% |
> | 123 | -5.96 | -6.59 | -7.45 | -7.70 | -8.83 | -8.97 | 74.0\% | 91.2\% | 0.48 | 0.45 | 0.65 | 0.65 | 0.60 | 0.62 | 54.4% |
> | 2023 | -5.73 | -6.67 | -7.31 | -7.57 | -8.83 | -8.87 | 69.6% | 89.8% | 0.48 | 0.46 | 0.66 | 0.65 | 0.60 | 0.61 | 50.5% |
>
> We can see that the results are very stable across different random seeds.

---

> ### Author Response · Authors · 2023-11-22
> **Gentle Reminder**
>
> Dear Reviewer SQ7r,
>
> Thank you for the time and effort you have put into evaluating our submission!
>
> In our response, we have further clarified the motivation, novelty, and contribution of our method, and done additional experiments. Besides, notably, we have provided extensive implementation details in Appendix A in the revision.
>
> We kindly remind you that the discussion phase is coming to the end. Please let us know if there are additional concerns we can address for you. If we have properly addressed your concerns, we sincerely hope you can reconsider the evaluation of our submission.

---

> > ### Comment · Reviewer_SQ7r · 2023-11-22
> >
> > Thanks for providing additional details! They are quite valuable.
> >
> > I observe that the experiment can be replicated using various random seeds.
> >
> > According to the new results and revised manuscript, I'd like to modify the score.

---

> > > ### Author Response · Authors · 2023-11-22
> > > **Thanks for your positive feedback!**
> > >
> > > Thank you agian for your positive feedback and kind support!
> > >
> > > We sincerely appreciate your recognition of our efforts and contribution.

---

### Official Review · Reviewer_eZ5C · 2023-11-01

**Soundness:** 3 good
**Presentation:** 3 good
**Contribution:** 3 good
**Rating:** 8
**Confidence:** 3

**Summary:**

This paper introduces a controllable and decomposed diffusion model for de novo molecular optimization. More specifically, the paper presented a method for molecular optimization while maintaining the constraint that the crucial binding region undergoes minimal alteration. The author conducts a comprehensive benchmarking of their work against numerous existing baselines, revealing strong performance across a wide range of quality-related metrics, although it does exhibit a lower score in terms of diversity.

**Strengths:**

1. The motivation is evident and firmly grounded.
2. The paper provides a great tool for chemistries to design the molecular and it is expected to be highly practical and beneficial.

**Weaknesses:**

I observed that you employed the reference ligand as the input for optimization, while most of the baseline methods, except RGA, did not. Moreover, the reference ligand exhibits exceptionally high quality, surpassing all the Gen-based baselines in terms of success rate. This leads me to question whether this comparison might be somewhat unfair to these baseline methods. I acknowledge that the primary focus of the method lies in optimization, making this kind of bias understandable. In light of this, I wonder whether it is necessary to include the Gen-based methods as baselines.

**Questions:**

1. How similar is your generated ligand to the reference ligand?
2. Can you demonstrate the influence of the reference ligand's quality on performance?

---

> ### Author Response · Authors · 2023-11-20
> **Response to Reviewer eZ5C**
>
> Thank you for your encouraging feedback. Please see below for our responses to the comments.
>
> **Q1: The reference ligand is used as the input for optimization.**
>
> A1: Thank you for pointing out the potential confusion about the initialization of conditional arm lists in our method. To clarify, our method initializes the conditional arm lists using DecompDiff, and not directly from reference ligands. We have clarified this point in the revision.
>
> **Q2: "How similar is your generated ligand to the reference ligand?"**
>
> A2: We believe there might be a misunderstanding here. We never used reference ligand as the input to our model. In all experiments, for each pocket, we start with the 20 ligands generated by DecompDiff, and then apply DecompOpt for 30 iterations. We have clarified this in the revision.
>
> We additionally test the **Novelty** and **Similarity** of generated ligands compared with the reference ligand. Novelty is defined as the ratio of generated ligands that are different from the reference ligands of the corresponding pockets in the test set. Similarity is defined as the Tanimoto Similarity between the generated ligands and the corresponding reference ligands. The results show that the generated ligands are not similar to reference ligands in the test set.
>
> Besides, we also test **Uniqueness** and **Diversity** of generated ligands. Uniqueness is the percentage of unique molecules among all the generated molecules. Diversity is the same as that in Section 4.1.
>
> |  | Novelty | Similarity | Uniqueness | Diversity |
> | --- | :---: | :---: | :---: | :---: |
> | LiGAN | 100% | 0.22 | 87.28% | 0.66 |
> | AR | 100% | 0.24 | 100% | 0.70 |
> | Pocket2Mol | 100% | 0.26 | 100% | 0.69 |
> | TargetDiff | 100% | 0.30 | 99.63% | 0.72 |
> | DecompDiff | 100% | 0.34 | 99.99% | 0.68 |
> | RGA | 100% | 0.37 | 96.82% | 0.41 |
> | DecompOpt | 100% | 0.36 | 100% | 0.60 |
>
> These results show that DecompOpt can design novel ligands, which is an important ability for drug discovery.
>
> **Q3: The influence of the reference ligand's quality on performance.**
>
> A3: We appreciate your concern about the potential influence of the initial ligands' quality on the performance of our optimization algorithm. Recognizing the importance of this issue, we have conducted an ablation study focusing on Vina Min score optimization, using ligands with high and low Vina Min scores as initializations for the arm lists.
> Due to limited resources, we chose to conduct this study on the protein 2V3R, which is randomly chosen from our test set. We generated 100 ligands using DecompDiff and selected 20 ligands with the highest and lowest Vina Min scores. These ligands were then used as the initial conditions for the optimization process.
>
> |  | High Vina min (Avg.) | High Vina min (Med.) | Low Vina min (Avg.) | Low Vina min (Med.) | $\Delta$ (high - low) (Avg.) |
> | --- | --- | --- | --- | --- | --- |
> | initial reference ligands | -8.54 | -8.48 | -7.08 | -7.04 | -1.46 |
> | DecompOpt | -9.12 | -8.96 | -9.00 | -8.96 | -0.12 |
>
> As shown in the experiment result, the optimization outcomes are slightly influenced by the quality of the initial ligands. However, regardless the quality of the initial ligands, DecompOpt can consistently improve the quality of the generated ligands.

---

> > ### Comment · Reviewer_eZ5C · 2023-11-21
> >
> > Thanks for providing additional details! They are quite valuable.
> >
> > Misunderstanding of reference ligand: thanks for pointing out that you didn't use reference ligand as input of the optimization process. It would be great to replace "reference ligand M" with a different name in algorithm 1 to alleviate confusion.
> >
> > According to the new results and revised manuscript, I'd like to modify the score.

---

> > > ### Author Response · Authors · 2023-11-21
> > > **Thank you for your positive feedback and support!**
> > >
> > > Thank you for your valuable suggestions and kind support! We will change the name in Algorithm 1 as you suggest.

---

### Official Review · Reviewer_Tgrc · 2023-11-01

**Soundness:** 3 good
**Presentation:** 2 fair
**Contribution:** 2 fair
**Rating:** 6
**Confidence:** 3

**Summary:**

Through this paper, the authors aim to establish a structure-based molecular optimization model that can be applied to both *de novo* molecular design and controllable generation. To accomplish this, the authors propose to train a diffusion model with decomposed ligands.

**Strengths:**

- The writing is easy to follow. The concept figure also aids the understanding.
- The experimental results show the effectiveness of the proposed method and ablation studies were also conducted. The experiments about controllability (R-group optimization and scaffold hopping) seem interesting.

**Weaknesses:**

- The authors did not provide the codebase to reproduce the results.
- The authors used the term *controllable generation* to indicate tasks like R-group design and scaffold hopping. However, *controllable* is a widely used expression with a general meaning and *de novo* molecular design can also be controllable in a broad sense. There is some lack of rigor in the choice of wording.

**Questions:**

- What is the specific difference between DecompOpt and DecompDiff? One of the claimed contributions is that the authors designed a unified framework via a decomposed diffusion model (the 3rd bullet point in the introduction), but DecompDiff has already proposed and applied this to SBDD.

---

> ### Author Response · Authors · 2023-11-20
> **Response to Reviewer Tgrc**
>
> Thank you for your positive feedback. Please see below for our responses to the comments.
>
> **Q1: Reproducibility.**
>
> A1: We are committed to releasing the codes, model checkpoints, and samples generated by our models upon acceptance. We hope these will contribute to the community. Besides, we have provided more implementation details in Appendix A in the revision.
>
> **Q2: The concept of controllable generation can be used not only in R-group optimization and scaffold hopping but also _de novo_ design. Lack of rigor in using the word.**
>
> A2: Thanks for pointing this out! We use _controllable generation_ here to refer to molecular generation with reference structures or substructures. In this sense, _controllable generation_ is distinguished from _de novo_ design. If the reviewer has any better suggestion on appropriate terminology, we would like to follow it.
>
> **Q3: Specific differences between DecompOpt and DecompDiff.**
>
> A3: Thanks for your question. DecompDiff incorporates decomposition into diffusion models for SBDD by introducing the decomposed prior, which only contains spatial information. In contrast, DecompOpt introduces reference arms as control signals to leverage both chemical and spatial semantics from discovered molecular substructures. To achieve this, based on DecompDiff, we introduce an SE(3)-invariant encoder to encode a reference arm and its surrounding subpocket as the condition. This makes DecompOpt fully utilize the advantages of decomposition. Moreover, thanks to controllability, we can effectively achieve molecular optimization and controllable molecular generation based on the model, while DecompDiff does not support.

---

### Meta-Review · Area_Chair_MUsW · 2023-12-08

**Metareview:**

This research focuses on the challenges faced in structure-based drug design using 3D generative models. While these models have shown promising results in generating ligands (molecules that bind to other specific molecules) given target binding sites, they struggle to design novel ligands with desired properties like high binding affinity, easy synthesizability, etc., particularly when the training data does not align with these desired properties. Additionally, existing methods are often focused on de novo design tasks, and lack flexibility for tasks requiring more control, such as R-group optimization and scaffold hopping.

To address these challenges, the authors present DecompOpt, a structure-based molecular optimization method based on a controllable and decomposed diffusion model. This new generation paradigm combines optimization with conditional diffusion models to create molecules with desired properties while adhering to the molecular grammar. DecompOpt also provides a unified framework for both de novo design and controllable generation, by decomposing ligands into substructures which allow for fine-grained control and local optimization.

Experiments show that DecompOpt can efficiently generate molecules with improved properties compared to existing de novo baselines, and exhibits strong potential for tasks requiring controllable generation. This research presents a novel advancement in structure-based drug design, offering both flexibility and control in generating molecules with desired properties.

The authors promised to make the reproducible codes and models available to the public upon acceptance, and carefully addressed the questions raised by reviewers. So the paper could be accepted if the authors keep their promise and includes these discussions in the final version.

**Justification For Why Not Higher Score:**

The majority of reviewers suggest acceptance above the borderline. One reviewer thinks the current draft good.

**Justification For Why Not Lower Score:**

One reviewer thinks the paper slightly below the acceptance borderline, whose questions are carefully addressed by the authors in rebuttal. This reviewer did not put further response after the rebuttal. So the majority opinion is followed.

---

### Decision · Program_Chairs · 2024-01-16

Accept (poster)